# DIME: Diffusion-Based Maximum Entropy Reinforcement Learning

**Onur Celik** [1]  **Zechu Li** [2]  **Denis Blessing** [1]  **Ge Li** [1]  **Daniel Palenicek** [3 4]  **Jan Peters** [3 4 5 6]  **Georgia Chalvatzaki** [2 4]
**Gerhard Neumann** [1]

## Abstract

Maximum entropy reinforcement learning (MaxEnt-RL) has become the standard approach to RL due to its beneficial exploration properties. Traditionally, policies are parameterized using Gaussian distributions, which significantly limits their representational capacity. Diffusion-based policies offer a more expressive alternative, yet integrating them into MaxEnt-RL poses challenges—primarily due to the intractability of computing their marginal entropy. To overcome this, we propose Diffusion-Based Maximum Entropy RL (DIME). *DIME* leverages recent advances in approximate inference with diffusion models to derive a lower bound on the maximum entropy objective. Additionally, we propose a policy iteration scheme that provably converges to the optimal diffusion policy. Our method enables the use of expressive diffusion-based policies while retaining the principled exploration benefits of MaxEnt-RL, significantly outperforming other diffusion-based methods on challenging high-dimensional control benchmarks. It is also competitive with state-of-the-art non-diffusion based RL methods while requiring fewer algorithmic design choices and smaller update-to-data ratios, reducing computational complexity[1].

## 1. Introduction

The maximum entropy reinforcement learning (MaxEnt-RL) objective augments the task reward in each time step with the entropy of the policy (Ziebart et al., 2008; Toussaint, 2009; Haarnoja et al., 2017; 2018b). This objective has

several favorable properties among which improved exploration (Ziebart, 2010; Haarnoja et al., 2017) is crucial for RL. Recent successful model-free RL algorithms leverage these favorable properties and build upon this framework (Bhatt et al., 2024; Nauman et al., 2024) improving sample efficiency and leading to remarkable results. However, the policies are traditionally parameterized using Gaussian distributions, significantly limiting their representational capacity. On the other hand, diffusion models (Sohl-Dickstein et al., 2015; Ho et al., 2020; Song et al., 2021; Karras et al., 2022) are highly expressive generative models and have proven beneficial in representing complex behavior policies (Reuss et al., 2023; Chi et al., 2023). However, important metrics such as the marginal entropy are intractable to compute (Zhou et al., 2024) which restricts their usage in RL. Because of this shortcoming, recent methods propose different ways to train diffusion-based methods in off-policy RL. While these methods are discussed in more detail in the related work section, most of them require additional techniques to add artificial (in most cases Gaussian) noise to the generated actions to induce exploration in the behavior generation process. Hence, they do not leverage the diffusion model to generate potentially non-Gaussian exploration patterns but fall back to mainly Gaussian exploration. Nonetheless, there have been significant advances in training diffusion-based models for approximate inference (Berner et al.; Richter & Berner). Since the policy improvement in MaxEnt-RL can also be cast as an approximate inference problem to the energy-based policy (Haarnoja et al., 2017), it is a natural step to explore these parallels.

We propose Diffusion-Based Maximum Entropy Reinforcement Learning (DIME). *DIME* leverages recent advances in approximate inference with diffusion models (Richter & Berner) to derive a lower bound on the MaxEnt objective. We propose a policy iteration framework with monotonic policy improvement that converges to the optimal diffusion policy. Additionally, building on recent off-policy RL algorithms such as Cross-Q (Bhatt et al., 2024) and distributional RL (Bellemare et al., 2017), we propose a practical version of DIME that can be used for training diffusion-based RL policies. On 13 challenging continuous high-dimensional control benchmarks, we empirically validate that DIME significantly outperforms other diffusion-based baselines on

---

[1]Autonomous Learning Robots, KIT [2]Interactive Robot Perception & Learning, TU Darmstadt [3]Intelligent Autonomous Systems, TU Darmstadt [4]Hessian.AI [5]German Research Center for AI [6]Centre for Cognitivie Science, TU Darmstadt . Correspondence to: Onur Celik <celik@kit.edu>.

*Proceedings of the 42$^{nd}$ International Conference on Machine Learning*, Vancouver, Canada. PMLR 267, 2025. Copyright 2025 by the author(s).

[1]https://alrhub.github.io/dime-website/

all environments and consistently outperforms other state-of-the-art RL methods based on a Gaussian policy on 10 out of 13 environments, while being computationally more efficient and requiring less algorithmic design choices as the current state of the art baseline BRO (Nauman et al., 2024).

## 2. Related Work

**Maximum Entropy RL.** The maximum entropy RL framework uses the entropy of the policy at each time step as an additional objective, providing a principled way of inducing exploration in the RL policy. It is different from entropy regularized RL (Neu et al., 2017), where the entropy of the policy is maximized only for the current time step. Haarnoja et al. (2017) proposed Soft-Q Learning, where amortized Stein variational gradient descent (Wang & Liu, 2016) (SVGD) is used to train a parameterized sampler that can sample from the energy-based policy. SAC (Haarnoja et al., 2018b) proposes an actor-critic RL method but frames the policy update as an approximate inference problem to the energy-based policy using a Gaussian policy parameterization. SAC has been extended to energy-based policies using SVGD in (Messaoud et al.), where the authors also propose a new method to estimate the entropy in closed form. While SVGD is a powerful method for learning an energy-based policy, it is harder to scale these approaches to high-dimensional control problems. For improving exploration, LSAC (Ishfaq et al., 2025) proposes leveraging Langevin Monte Carlo (Welling & Teh, 2011) in conjunction with a distributed critic objective to sample a state-action value. Haarnoja et al. (2018a) proposes learning a latent variable model as a policy representation, but relies on the change of variable formula to express the density of the policy by calculating the Jacobian of the transformations. Recent advances of SAC also define the state-of-the-art in off-policy RL in many domains, such as CrossQ (Bhatt et al., 2024) and BRO (Nauman et al., 2024). CrossQ proposed removing the target network by leveraging batch renormalization and BRO scales to large networks in RL by using several methods such as optimistic exploration (Nauman & Cygan, 2023), network resets (Nikishin et al., 2022), weight decay, and high update-to-data ratios.

**Diffusion-Based Policies in RL.** Early works have researched diffusion models in offline RL (Lange et al., 2012; Levine et al., 2020) as trajectory generators (Janner et al., 2022) or as expressive policy representations (Wang et al., 2023; Kang et al., 2023; Hansen-Estruch et al., 2023; Chen et al., 2023; Ding & Jin, 2024; Mao et al., 2024; Fang et al., 2025; Lu et al., 2023). More recently, diffusion models in online RL have become more popular. DIPO (Yang et al., 2023) proposes training a diffusion-based policy using a behavior cloning loss. The actions in the replay buffer serve as target actions for the policy improvement step and are

updated using the gradients of the Q-function $\nabla_a Q(s, a)$. DIPO has been extended to develop methods for learning multi-modal behaviors(Li et al., 2024) by leveraging hierarchical clustering to isolate different behavior modes. DIPO relies on the stochasticity inherent to the diffusion model for exploration and does not explicitly control it via an objective. QSM (Psenka et al., 2024) directly matches the policy's score with the gradient of the Q-function $\nabla_a Q(s, a)$. While their objective avoids differentiating through the whole diffusion chain, the proposed objective disregards the entropy of the policy and, therefore, exploration. Consequently, QSM needs to add noise to the final action of the diffusion chain. More recently, DACER (Wang et al., 2024) proposed using the data-generating process as the policy representation and backpropagating the gradients through the diffusion chain. However, they do not consider a backward process as we do, and their objective for updating the diffusion model is based on the expected Q-values only. To incentivize the exploration, DACER adds diagonal Gaussian noise to the sampled actions, where the variance of this noise is controlled by a scaling term that is updated automatically using an approximation of the marginal entropy by extracting a Gaussian Mixture Model from the diffusion policy. Concurrently, QVPO (Ding et al., 2024) proposed weighting their diffusion loss with their respective Q-values after applying transformations. However, QVPO relies on sampling actions from a uniform distribution to enforce exploration.

DIME distinguishes from prior works in that we use the maximum entropy RL framework for training the diffusion policy, which was not considered before. This allows direct control of the exploration-exploitation trade-off arising naturally through this objective without the need for additional approximations. DIME is leveraging the diffusion model to generate non-Gaussian exploration actions which is in contrast to most other diffusion RL approaches that still require including Gaussian or uniform exploration noise.

**Approximate Inference with Diffusion Models.** Early works on approximate inference with diffusion models were formalized as a stochastic optimal control problem using Schrödinger-Föllmer diffusions (Dai Pra, 1991; Tzen & Raginsky, 2019; Huang et al., 2021) and only recently realized with deep-learning based approaches (Vargas et al., 2023; Zhang & Chen, 2021). Vargas et al.; Berner et al. later extended these results to denoising diffusion models. A more general framework where both forward and backward processes of the diffusion model are learnable was concurrently proposed by Richter & Berner; Nusken et al. (2024). Recently, many extensions have been proposed, see e.g. (Akhound-Sadegh et al., 2024; Noble et al., 2024; Geffner & Domke, 2023; Zhang et al., 2023; Chen et al., 2024; Blessing et al., 2025b;a; Chen et al., 2025). Our work can be seen as an instance of the sampler presented in (Berner et al.). However, our formulation allows using different

diffusion samplers such as those presented in (Richter & Berner; Blessing et al., 2025a), while we restrict ourselves in this work to the sampler presented in (Berner et al.).

## 3. Preliminaries

### 3.1. Maximum Entropy Reinforcement Learning

**Notation** We consider the task of learning a policy $\pi : \mathcal{S} \times \mathcal{A} \to \mathbb{R}^+$, where $\mathcal{S}$ and $\mathcal{A}$ denote a continuous state and action space, respectively using reinforcement learning (RL). We formalize the RL problem using an infinite horizon Markov decision process consisting of the tuple $(\mathcal{S}, \mathcal{A}, r, p, \rho_\pi, \gamma)$, with bounded reward function $r : \mathcal{S} \times \mathcal{A} \to [r_{\min}, r_{\max}]$ and transition density $p : \mathcal{S} \times \mathcal{S} \times \mathcal{A} \to \mathbb{R}^+$ which denotes the likelihood for transitioning into a state $s' \in \mathcal{S}$ when being in $s \in \mathcal{S}$ and executing an action $a \in \mathcal{A}$. We follow (Haarnoja et al., 2018b) and slightly overload $\rho_\pi$ which denotes the state and state-action marginals induced by a policy $\pi$. Moreover, $\gamma \in [0, 1)$ denotes the discount factor. For brevity, we use $r_t \triangleq r(s_t, a_t)$. Lastly, we denote objective functions that we aim to maximize as $J$ and minimize as $\mathcal{L}$.

**Control as inference.** The goal of maximum entropy reinforcement learning (MaxEnt-RL) is to jointly maximize the sum of expected rewards and entropies of a policy

$$J(\pi) = \sum_{t=l}^{\infty} \gamma^{t-l} \mathbb{E}_{\rho_\pi} \left[ r_t + \alpha \mathcal{H}(\pi(a_t|s_t)) \right], \quad (1)$$

where $\mathcal{H}(\pi(a|s)) = -\int \pi(a|s) \log \pi(a|s) \mathrm{d}a$ is the differential entropy, and $\alpha \in \mathbb{R}^+$ controls the exploration exploitation trade-off (Haarnoja et al., 2017). To keep the notation uncluttered we absorb $\alpha$ into the reward function via $r \leftarrow r/\alpha$. Defining the $Q$-function of a policy $\pi$ as

$$Q^\pi(s_t, a_t) = r_t + \sum_{l=1}^{\infty} \gamma^l \mathbb{E}_{\rho_\pi} \left[ r_{t+l} + \mathcal{H}\left( \pi(a_{t+l}|s_{t+l}) \right) \right], \quad (2)$$

with $Q^\pi : \mathcal{S} \times \mathcal{A} \to \mathbb{R}$, the MaxEnt objective can be cast as an approximate inference problem of the form

$$\mathcal{L}(\pi) = D_{\mathrm{KL}} \left( \pi(a_t|s_t) \Big| \frac{\exp Q^\pi(s_t, a_t)}{\mathcal{Z}^\pi(s_t)} \right), \quad (3)$$

in a sense that $\max_\pi J(\pi) = \min_\pi \mathcal{L}(\pi)$. Here, $D_{\mathrm{KL}}$ denotes the Kullback-Leibler divergence and

$$\mathcal{Z}^\pi(s) = \int \exp Q^\pi(s, a) \mathrm{d}a \quad (4)$$

is the state-dependent normalization constant.

**Policy iteration** is a two-step iterative update scheme that is, under certain assumptions, guaranteed to converge to the

optimal policy with respect to the maximum entropy objective. The two steps include policy evaluation and policy improvement. Given a policy $\pi$, policy evaluation aims to evaluate the value of $\pi$. To that end, (Haarnoja et al., 2018b) showed that repeated application of the Bellman backup operator $\mathcal{T}^\pi Q^k$ with

$$\mathcal{T}^\pi Q(s_t, a_t) \triangleq r_t + \gamma \mathbb{E}\left[Q(s_{t+1}, a_{t+1}) + \mathcal{H}(a_{t+1}|s_{t+1})\right], \quad (5)$$

converges to $Q^\pi$ as $k \to \infty$, starting from any $Q$. To update the policy, that is, to perform the policy improvement step, the $Q$-function of the previous evaluation step, $Q^{\pi_{\mathrm{old}}}$ is used to obtain a new policy according to

$$\pi_{\mathrm{new}} = \arg\min_{\pi \in \Pi} D_{\mathrm{KL}} \left( \pi(a_t|s_t) \Big| \frac{\exp Q^{\pi_{\mathrm{old}}}(s_t, a_t)}{\mathcal{Z}^{\pi_{\mathrm{old}}}(s_t)} \right), \quad (6)$$

where $\Pi$ is a set of policies such as a family of parameterized distributions. Note that $\mathcal{Z}^{\pi_{\mathrm{old}}}(s_t)$ is not required for optimization as it is independent of $\pi$. Haarnoja et al. (2018b) showed that for all state-action pairs $(s, a) \in \mathcal{S} \times \mathcal{A}$ it holds that $Q^{\pi_{\mathrm{new}}}(s, a) \geq Q^{\pi_{\mathrm{old}}}(s, a)$ ensuring that policy iteration converges to the optimal policy $\pi^*$ in the limit of infinite repetitions of policy evaluation and improvement.

### 3.2. Denoising Diffusion Policies

For a given state $s \in \mathcal{S}$, we consider a stochastic process on the time-interval $[0, T]$ given by an Ornstein-Uhlenbeck (OU) process [2] (Särkkä & Solin, 2019)

$$\mathrm{d}a_t = -\beta_t a_t \mathrm{d}t + \eta \sqrt{2\beta_t} \mathrm{d}B_t, \quad a_0 \sim \vec{\pi}_0(\cdot|s), \quad (7)$$

with diffusion coefficient $\beta : [0, T] \to \mathbb{R}^+$, standard Brownian motion $(B_t)_{t \in [0,T]}$, and some target policy $\vec{\pi}_0$. For $t, l \in [0, T]$, we denote the marginal density of Eq. 7 at $t$ as $\vec{\pi}_t$ and the conditional density at time $t$ given $l$ as $\vec{\pi}_{t|l}$. Eq. 7 is commonly referred to as *forward* or *noising process* since, for a suitable choice of $\beta$, it holds that $\vec{\pi}_T \approx \mathcal{N}(0, \eta^2 I)$. Denoising diffusion models leverage the fact, that the time-reversed process of Eq. 7 is given by

$$\mathrm{d}a_t = \left(-\beta_t a_t \mathrm{d}t - 2\eta^2 \beta_t \nabla \log \vec{\pi}_t(a_t|s)\right) + \eta \sqrt{2\beta_t} \mathrm{d}B_t, \quad (8)$$

starting from $\overleftarrow{\pi}_T = \vec{\pi}_T \approx \mathcal{N}(0, \eta^2 I)$ and running backwards in time (Nelson, 2020; Anderson, 1982; Haussmann & Pardoux, 1986). For the *backward*, *generative* or *denoising process* (Eq. 8), we denote the density as $\overleftarrow{\pi}$. Here, time-reversal means that the marginal densities align, i.e., $\vec{\pi}_t = \overleftarrow{\pi}_t$ for all $t \in [0, T]$. Hence, starting from $a_T \sim \mathcal{N}(0, \eta^2 I)$, one can sample from the target policy $\vec{\pi}_0$ by simulating Eq. 8. However, for most densities $\vec{\pi}_0$,

---

[2]Please note, for clarity, we slightly abuse notation by using $t$ to denote the time in the stochastic process. This should not be confused with the time step in RL. The distinction becomes clear when we discretize the processes.

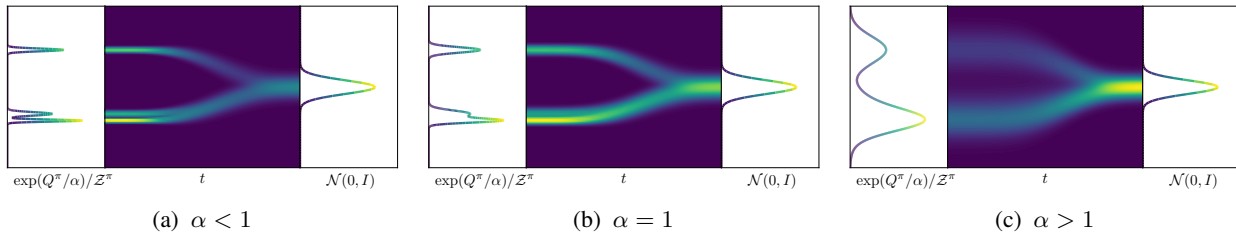

(a) $\alpha < 1$        (b) $\alpha = 1$        (c) $\alpha > 1$

*Figure 1.* **The effect of the reward scaling parameter** $\alpha$. The figures in (a)-(b) show diffusion processes for different $\alpha$ values starting at a prior distribution $\mathcal{N}(0, I)$ and going backward in time to approximate the target distribution $\exp(Q^\pi/\alpha)/Z^\pi$. Small values for $\alpha$ (a) lead to concentrated target distributions with less noise in the diffusion trajectories especially at the last time steps. The higher $\alpha$ becomes (b) and (c), the more the target distribution is smoothed and the distribution of the samples at the last time steps becomes more noisy. Therefore, the parameter $\alpha$ directly controls the exploration by enforcing noisier samples the higher $\alpha$ becomes.

the scores $(\nabla \log \vec{\pi}_t(a_t|s))_{t \in [0,T]}$ are intractable, requiring numerical approximations. To address this, denoising score-matching objectives are commonly employed, that is,

$$\mathcal{L}_{\text{SM}}(\theta) = \mathbb{E}\left[\beta_t \| f_t^\theta(a_t, s) - \nabla \log \vec{\pi}_{t|0}(a_t|a_0, s)\|^2\right], \tag{9}$$

where $t$ is sampled on $[0, T]$ and $f^\theta$ denotes a parameterized score network (Hyvärinen & Dayan, 2005; Vincent, 2011). For OU processes, the conditional densities $\nabla \log \vec{\pi}_{t|0}$ are explicitly computable, making the objective tractable for optimizing $\theta$ (Song et al., 2021). Once trained, the score network $f^\theta$ can be used to simulate the denoising process

$$\mathrm{d}a_t = \left(-\beta_t a_t \mathrm{d}t - 2\eta^2 \beta_t f_t^\theta(a_t, s)\right) + \eta\sqrt{2\beta_t}\mathrm{d}B_t, \tag{10}$$

to obtain samples $a_0 \sim \pi_0^\theta$ that are approximately distributed according to $\vec{\pi}_0$. Here, $\pi_t^\theta$ denotes the marginal distribution of Eq. 10 at time $t$. While score-matching techniques work well in practice, they cannot be applied to maximum entropy reinforcement learning. This is because the expectation in Eq. 9 requires samples $a_0 \sim \vec{\pi}_0 \propto \exp Q^\pi$ which are not available. However, in the next section, we build on recent advances in approximate inference to optimize diffusion models without requiring samples from $a_0$, relying instead on evaluations of $Q^\pi$.

## 4. Diffusion-Based Maximum Entropy RL

Here, we explain how diffusion models can be used within a maximum entropy RL framework. To that end, we express the maximum entropy objective as an approximate inference problem for diffusion models. We then use these results to introduce a policy iteration scheme that provably converges to the optimal policy. Lastly, we propose a practical algorithm for optimizing diffusion models.

### 4.1. Control as Inference for Diffusion Policies

Directly maximizing the maximum entropy objective

$$J(\overleftarrow{\pi}) = \sum_{t=l}^{\infty} \gamma^{t-l} \mathbb{E}_{\rho_\pi}\left[r_t(s_t, a_t^0) + \alpha\mathcal{H}(\overleftarrow{\pi}_0(a_t^0|s_t))\right],$$

for a diffusion model is difficult as the marginal entropy $\mathcal{H}(\overleftarrow{\pi}_0(a|s))$ of the denoising process in Eq. 8 is intractable. Please note that we use superscripts for the actions to indicate the diffusion step to avoid collisions with the time step used in RL. Moreover, we will again absorb $\alpha$ into the reward and use $r_t \triangleq r(s_t, a_t^0)$. To overcome this intractability, we propose to maximize a lower bound. We start by discretizing the stochastic processes introduced in Section 3.2 and use the results as a foundation to derive this lower bound. Note that while similar results can be derived from a continuous-time perspective (see e.g., Berner et al.; Richter & Berner; Nusken et al. (2024)), such derivation would require a background in stochastic calculus, making it less accessible to a broader audience.

The Euler-Maruyama (EM) discretization (Särkkä & Solin, 2019) of the noising (Eq. 7) and denoising (Eq. 8) process is given by

$$a^{n+1} = a^n - \beta_n a^n \delta + \epsilon_n \quad \text{and} \tag{11}$$

$$a^{n-1} = a^n + \left(\beta_n a^n + 2\eta^2 \beta_n \nabla \log \vec{\pi}_n(a^n|s)\right)\delta + \xi_n, \tag{12}$$

respectively, with $\epsilon_n, \xi_n \sim \mathcal{N}(0, 2\eta^2 \beta_n \delta I)$. Here, $\delta$ denotes a constant discretization step size such that $N = T/\delta$ is an integer. To simplify notation, we write $a^n$, instead of $a^{n\delta}$. Under the EM discretization, the noising and denoising process admit the following joint distributions

$$\vec{\pi}_{0:N}(a^{0:N}|s) = \vec{\pi}_0(a^0|s)\prod_{n=0}^{N-1}\vec{\pi}_{n+1|n}(a^{n+1}|a^n, s), \tag{13}$$

$$\overleftarrow{\pi}_{0:N}(a^{0:N}|s) = \overleftarrow{\pi}_N(a^N|s)\prod_{n=1}^{N}\overleftarrow{\pi}_{n-1|n}(a^{n-1}|a^n, s), \tag{14}$$

in a sense that $\vec{\pi}_{0:N}$ and $\overleftarrow{\pi}_{0:N}$ converge to the law of $(a_t)_{t \in [0,T]}$ in Eq. 7 and 8, as $\delta \to 0$, respectively (Doucet et al., 2022). Here, $\vec{\pi}_{n+1|n}$ and $\overleftarrow{\pi}_{n-1|n}$ are Gaussian transition densities that directly follow from Eq. 11 and 12.

To obtain a maximum entropy objective for diffusion models, we make use of the following lower bound on the marginal entropy, that is, $\mathcal{H}(\overleftarrow{\pi}_0(a_0|s)) \geq \ell_{\overleftarrow{\pi}}(a^0, s)$, where

$$\ell_{\overleftarrow{\pi}}(a^0, s) = \mathbb{E}_{\overleftarrow{\pi}_{0:N}}\left[\log \frac{\overleftarrow{\pi}_{1:N|0}(a^{1:N}|a^0, s)}{\overleftarrow{\pi}_{0:N}(a^{0:N}|s)}\right]. \quad (15)$$

Please note that similar bounds have been used, e.g., in (Agakov & Barber, 2004; Tran et al., 2015; Ranganath et al., 2016; Maaløe et al., 2016; Arenz et al., 2018), or, more generally, follow from the data processing inequality (Cover, 1999). A derivation can be found in Appendix A. From Eq. 15, it directly follows that

$$J(\overleftarrow{\pi}) \geq \bar{J}(\overleftarrow{\pi}) = \sum_{t=l}^{\infty} \gamma^{t-l}\mathbb{E}_{\rho_\pi}\left[r_t + \ell_{\overleftarrow{\pi}}(a_t^0, s_t)\right]. \quad (16)$$

Next, we cast Eq. 16 as an approximate inference problem to make the objective more interpretable. To that end, let us define the $Q$-function of a denoising policy $\overleftarrow{\pi}$ with respect to the maximum entropy objective $\bar{J}$ as

$$Q^{\overleftarrow{\pi}}(s_t, a_t^0) = r_t + \sum_{l=1} \gamma^l \mathbb{E}_{\rho_\pi}\left[r_{t+l} + \ell_{\overleftarrow{\pi}}(a_{t+l}^0, s_{t+l})\right], \quad (17)$$

with $Q^{\overleftarrow{\pi}} : \mathcal{S} \times \mathcal{A} \to \mathbb{R}$. With Eq. 17 we identify the corresponding approximate inference problem as finding $\overleftarrow{\pi}$ which minimizes (please see Appendix A for derivation)

$$\bar{\mathcal{L}}(\overleftarrow{\pi}) = D_{\text{KL}}\left(\overleftarrow{\pi}_{0:N}(a^{0:N}|s)|\vec{\pi}_{0:N}(a^{0:N}|s)\right), \quad (18)$$

where the target policy, i.e., the marginal of the noising process in Eq. 13 is given by the exponentiated $Q$-function of the diffusion policy

$$\vec{\pi}_0(a^0|s) = \frac{\exp Q^{\overleftarrow{\pi}}(s, a^0)}{\mathcal{Z}^{\overleftarrow{\pi}}(s)}. \quad (19)$$

Recall from Section 3.2 that we aim to time-reverse the noising process, that is, to ensure for all states $s \in \mathcal{S}$, it holds that $\overleftarrow{\pi}_{0:N} = \vec{\pi}_{0:N}$. Please note that this is precisely what Eq. 18 is trying to accomplish, i.e., we aim to learn a diffusion model $\overleftarrow{\pi}$, such that the denoising process time-reverses the noising process, and, in particular, has a marginal distribution given by $\pi_0 = \exp Q^{\overleftarrow{\pi}}/\mathcal{Z}^{\overleftarrow{\pi}}$. Lastly, from the data processing inequality, it directly follows that

$$D_{\text{KL}}\left(\overleftarrow{\pi}_0(a^0|s)\Big|\frac{\exp Q^{\overleftarrow{\pi}}(s, a^0)}{\mathcal{Z}^{\overleftarrow{\pi}}(s)}\right)$$
$$\leq D_{\text{KL}}\left(\overleftarrow{\pi}(a^{0:N}|s)|\vec{\pi}(a^{0:N}|s)\right), \quad (20)$$

which shows the approximate inference problem in Eq. 18 indeed optimizes the same inference problem stated in Eq. 3. Next, we will use these results to develop a policy iteration scheme for diffusion models.

### 4.2. Diffusion-based Policy Iteration

We propose a policy iteration scheme for learning an optimal maximum entropy policy, similar to (Haarnoja et al., 2018b). However, here we restrict the family of stochastic actors to diffusion policies $\overleftarrow{\pi} \in \overleftarrow{\Pi} \subset \Pi$. Throughout this section, we assume finite action spaces to enable theoretical analysis, but relax this assumption in Section 4.3. All proofs of this section are deferred to Appendix A.

For policy evaluation, we aim to compute the value of a policy $\overleftarrow{\pi}$. We define the Bellman backup operator as

$$\mathcal{T}^{\overleftarrow{\pi}}Q(s_t, a_t^0) \triangleq r_t + \gamma\mathbb{E}\left[Q(s_{t+1}, a_{t+1}^0) + \ell_{\overleftarrow{\pi}}(a_{t+1}^0, s_{t+1})\right]. \quad (21)$$

Note that Eq. 21 contains the entropy-lower bound $\ell_{\overleftarrow{\pi}}$. By applying standard convergence results for policy evaluation (Sutton & Barto, 1999) we can obtain the value of a policy by repeatedly applying $\mathcal{T}^{\overleftarrow{\pi}}$ as established in Proposition 4.1.

**Proposition 4.1** (Policy Evaluation). *Let $\mathcal{T}^{\overleftarrow{\pi}}$ be the Bellman backup operator for a diffusion policy $\overleftarrow{\pi}$ as defined in Eq. 21. Further, let $Q^0 : \mathcal{S} \times \mathcal{A} \to \mathbb{R}$ and $Q^{k+1} = \mathcal{T}^{\overleftarrow{\pi}}Q^k$. Then, it holds that $\lim_{k\to\infty} Q^k = Q^{\overleftarrow{\pi}}$ where $Q^{\overleftarrow{\pi}}$ is the Q value of $\overleftarrow{\pi}$.*

For the policy improvement step, we seek to improve the current policy based on its value using the $Q$-function. Formally, we need to solve the approximate inference problem

$$\overleftarrow{\pi}^{\text{new}} = \arg\min_{\overleftarrow{\pi}\in\overleftarrow{\Pi}} D_{\text{KL}}\left(\overleftarrow{\pi}_{0:N}(a^{0:N}|s)|\vec{\pi}_{0:N}^{\text{old}}(a^{0:N}|s)\right), \quad (22)$$

for all $s \in \mathcal{S}$, where $\vec{\pi}_{0:N}^{\text{old}}(a^{0:N}|s)$ is as in Eq. 13 with marginal density

$$\vec{\pi}_0^{\text{old}}(a^0|s) = \frac{\exp Q^{\overleftarrow{\pi}_{\text{old}}}(s, a^0)}{\mathcal{Z}^{\overleftarrow{\pi}_{\text{old}}}(s)}. \quad (23)$$

Indeed, solving Eq. 22 results in a policy with higher value as established below.

**Proposition 4.2** (Policy Improvement). *Let $\overleftarrow{\pi}_{old}, \overleftarrow{\pi}_{new} \in \overleftarrow{\Pi}$ be defined as in Eq. 23 and 22, respectively. Then for all $(s, a) \in \mathcal{S} \times \mathcal{A}$ it holds that $Q^{\overleftarrow{\pi}_{new}}(s, a) \geq Q^{\overleftarrow{\pi}_{old}}(s, a)$.*

Combining these results leads to the policy iteration method which alternates between policy evaluation (Proposition 4.1) and policy improvement (Proposition 4.2) and provably converges to the optimal policy in $\overleftarrow{\Pi}$ (Proposition 4.3).

**Proposition 4.3** (Policy Iteration). *Let $\overleftarrow{\pi}^0, \overleftarrow{\pi}^{i+1}, \overleftarrow{\pi}^i, \overleftarrow{\pi}_* \in \overleftarrow{\Pi}$. Further, let $\overleftarrow{\pi}^{i+1}$ be the policy obtained from $\overleftarrow{\pi}^i$ after a policy evaluation and improvement step. Then, for any starting policy $\overleftarrow{\pi}^0$ it holds that $\lim_{i\to\infty} \overleftarrow{\pi}^i = \overleftarrow{\pi}^*$, with $\overleftarrow{\pi}^*$ such that for all $\overleftarrow{\pi} \in \overleftarrow{\Pi}$ and $(s, a) \in \mathcal{S} \times \mathcal{A}$ it holds that $Q^{\overleftarrow{\pi}^*}(s, a) \geq Q^{\overleftarrow{\pi}}(s, a)$.*

However, performing policy iteration until convergence is in practice often intractable, particularly for continuous control tasks. As such, we will introduce a practical algorithm next.

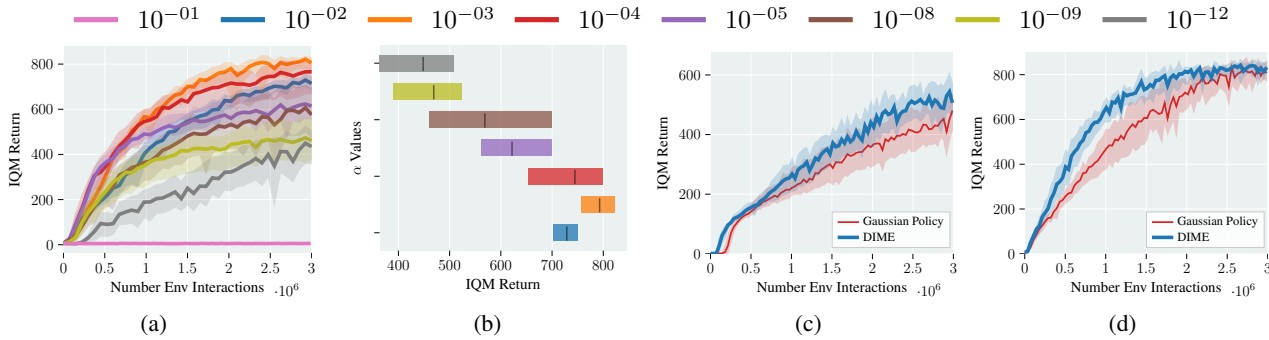

*Figure 2.* **Reward Scaling Sensitivity (a)-(b)**. The $\alpha$ parameter controls the exploration-exploitation trade-off. (a) shows the learning curves for varying values on DMC's dog-run task. Too high $\alpha$ values ($\alpha = 0.1$) do not incentivize learning whereas too small $\alpha$ values ($\alpha \leq 10^{-5}$) converge to suboptimal behavior. (b) shows the aggregated end performance for each learning curve in (a). For increasing $\alpha$ values, the end performance increases until it reaches an optimum at $\alpha = 10^{-3}$ after which the performance starts dropping. **Diffusion Policy Benefit (c) and (d)**. We compare DIME to a Gaussian policy with the same implementation details as DIME on the (a) humanoid-run and (b) dog-run tasks. The diffusion-based policy reaches a higher return (a) and converges faster.

### 4.3. DIME: A Practical Diffusion RL Algorithm

To obtain a practical algorithm, we use a parameterized function approximation for the $Q$-function and the policy, that is, $Q_\phi$ and $\pi^\theta$, with parameters $\phi$ and $\theta$, respectively. Here, $\pi^\theta$ is represented by a parameterized score network, see Eq. 10. To perform approximate policy evaluation, we can minimize the Bellman residual,

$$J_Q(\phi) = \frac{1}{2}\mathbb{E}\left[\left(Q_\phi(s_t, a_t^0) - Q_{\text{target}}(s_t, a_t^0)\right)^2\right], \quad (24)$$

using stochastic gradients with respect to $\phi$. We provide implementation details in Section 4.4. Moreover, the expectation is computed using state-action pairs collected from environment interactions and saved in a replay buffer. For policy improvement, we solve the approximate inference problem

$$\mathcal{L}(\theta) = D_{\text{KL}}\left(\pi_{0:N}^\theta(a^{0:N}|s)|\vec{\pi}_{0:N}(a^{0:N}|s)\right), \quad (25)$$

where the target policy, i.e., the marginal of the noising process in Eq. 13 is given by the approximate $Q$-function

$$\vec{\pi}_0(a^0|s) = \frac{\exp Q_\phi(s, a^0)}{\mathcal{Z}_\phi(s)}, \quad (26)$$

where states are again sampled from a replay buffer. Further expanding $\mathcal{L}(\theta)$ yields

$$\mathcal{L}(\theta) = \mathbb{E}_{\pi^\theta}\left[\log \pi_N^\theta(a^N|s) - Q_\phi(s, a^0)\right.$$
$$\left. + \sum_{n=1}^{N} \log \frac{\pi_{n-1|n}^\theta(a^{n-1}|a^n, s)}{\vec{\pi}_{n|n-1}(a^n|a^{n-1}, s)}\right] + \log \mathcal{Z}_\phi(s), \quad (27)$$

showing that $\mathcal{Z}_\phi$ is not needed to minimize Eq. 27 as it is independent of $\theta$. Moreover, contrary to the score-matching objective (see Eq. 9) that is commonly used to optimize

diffusion models, stochastic optimization of $\mathcal{L}(\theta)$ does not need access to samples $a_0 \sim \exp Q_\phi/\mathcal{Z}_\phi$, instead relying on stochastic gradients obtained via reparameterization trick (Kingma, 2013) using samples from the diffusion model $\pi^\theta$.

### 4.4. Implementation Details

**Autotuning Temperature.** We follow implementations like SAC (Haarnoja et al., 2018c) where the reward scaling parameter $\alpha$ (also see Fig. 1) is not absorbed into the reward but scales the entropy term. Choosing $\alpha$ depends on the reward ranges and the dimensionality of the action space, which requires tuning it per environment. We instead follow prior works (Haarnoja et al., 2018c) for auto-tuning $\alpha$ by optimizing

$$J(\alpha) = \alpha\left(\mathcal{H}_{\text{target}} - \ell_\mathcal{H}^\theta\right), \quad (28)$$

where $\mathcal{H}_{\text{target}}$ is a target value for the mismatch between the noising and denoising processes measured by the log ratio.

**Autotuning Diffusion Coefficient.** Please note that the objective function in Eq. 27 is fully differentiable with respect to parameters of the diffusion process. As such, we additionally treat the diffusion coefficient $\beta$ as a learnable parameter that is optimized end-to-end, further reducing the need for manual hyperparameter tuning. Further details on the parameterization can be found in Appendices D and G.

$Q$-**function.** Following Bhatt et al. (2024) we adopt the CrossQ algorithm, i.e., we use Batch Renormalization in the Q-function and avoid a target network for calculating $Q_{\text{target}}$. When updating the Q-function, the values for the current and next state-action pairs are queried in parallel. The next Q-values are used as target values where the gradients are stopped. Additionally, we employ distributional Q learning as proposed by (Bellemare et al., 2017). The details are described in Appendix D.

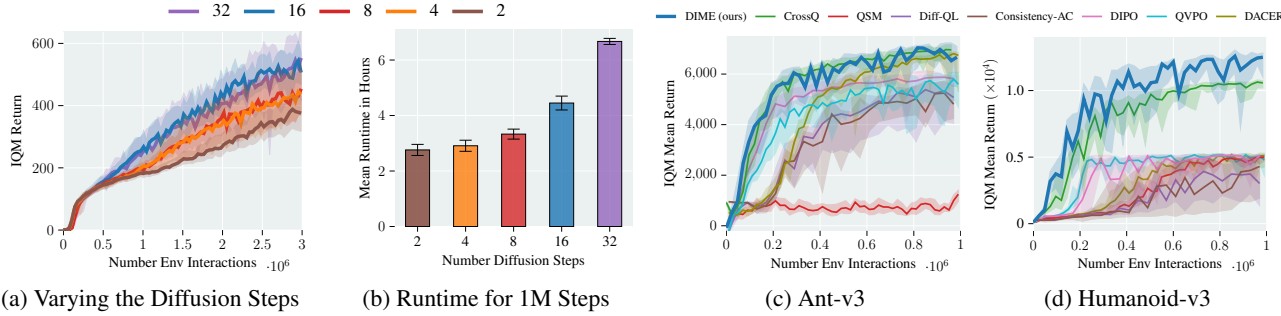

(a) Varying the Diffusion Steps      (b) Runtime for 1M Steps      (c) Ant-v3      (d) Humanoid-v3

*Figure 3.* **Varying the Number of diffusion steps (a)-(b).** The number of diffusion steps might affect the performance and the computation time. (a) shows DIME's learning curves for varying diffusion steps. *Two* diffusion steps perform badly, whereas *four* and *eight* diffusion steps perform similar but still worse than *16* and *32* diffusion steps which perform similarly. (b) shows the computation time for 1MIO steps of the corresponding learning curves. The smaller the diffusion steps, the less computation time is required. **Learning Curves on Gym Benchmark Suite (c)-(d).** We compare DIME against various diffusion baselines and CrossQ on the (c) *Ant-v3* and (d) *Humanoid-v3* from the Gym suite. While all diffusion-based methods are outperformed by DIME, DIME performs on par with CrossQ on the Ant environment. DIME performs favorably on the high-dimensional *Humanoid-v3* environment, where it also outperforms CrossQ.

# 5. Experiments

We analyze DIME's algorithmic features with an intensive ablation study where we clarify the role of the reward scaling parameter $\alpha$, the effect of varying diffusion steps, and the gained performance boost when using a diffusion policy representation over a Gaussian representation. Additional analysis on employing distributional Q learning is discussed in the Appendix G.

In a broad range of 13 sophisticated learning environments from different benchmark suits, ranging from mujoco gym (Brockman et al., 2016), deepmind control suit (DMC) (Tunyasuvunakool et al., 2020), and myo suite (Caggiano et al., 2022), we compare DIME's performance against state-of-the-art RL baselines that employ diffusion and Gaussian policy parameterizations. The considered environments are challenging locomotion and manipulation learning tasks with up to 39-dimensional action and 223-dimensional observation spaces.

We consider QSM (Psenka et al., 2024), Diffusion-QL (Wang et al., 2023), Consistency-AC (Ding & Jin, 2024), DIPO (Yang et al., 2023), QVPO (Ding et al., 2024), and DACER (Wang et al., 2024) as baselines for diffusion-based policy representations.

Additionally, we compare against the state-of-the-art RL methods CrossQ (Bhatt et al., 2024) and BRO (Nauman et al., 2024), where we have used the provided learning curves from the latter. Both methods use a Gaussian parameterized policy and have shown remarkable results.

We have run the learning curves for 10 seeds using the official code releases and report the *interquartile mean (IQM)* with a 95% stratified bootstrap confidence interval as suggested by Agarwal et al. (2021).

## 5.1. Ablation Studies

**Exploration Control.** The parameter $\alpha$ balances the exploration-exploitation trade-off by scaling the reward signal. We analyze the effect of this parameter by comparing DIME's learning curves with different $\alpha$ values on the dog-run task from the DMC (see Fig. 2a). Additionally, we show the performance of the last return measurements for each learning curve in Fig. 2b. Too high $\alpha$ values ($\alpha = 0.1$) do not incentivize maximizing the task's return, leading to no learning at all, whereas small values ($\alpha \leq 10^{-5}$) lead to suboptimal performance because the policy does not explore sufficiently. We can also see a clear trend that starting from $\alpha = 10^{-12}$, the performance gradually increases until the best performance is reached for $\alpha = 10^{-3}$.

**Diffusion Policy Benefit.** We aim to analyze the performance benefits of the diffusion-parameterized policy compared to a Gaussian parameterization in the same setup by only exchanging the policy and the corresponding policy update. This comparison ensures that the Gaussian policy is trained with the identical implementation details from DIME as described in Sec. 4.4 and showcases the performance benefits of a diffusion-based policy. Fig. 2c and 2d show the learning curves of both versions on DMC's humanoid-run and dog-run environments. The diffusion policy's expressivity leads to a higher aggregated return in the humanoid-run and to significantly faster convergence in the high-dimensional dog-run task. We attribute this performance benefit to an improved exploration behavior.

**Number of Diffusion Steps.** The number of diffusion steps determines how accurately the stochastic differential equations are simulated and is a hyperparameter that affects the performance. Usually, the higher the number of diffusion steps the better the model performs at the burden of higher computational costs. In Fig. 3a we plot DIME's perfor-

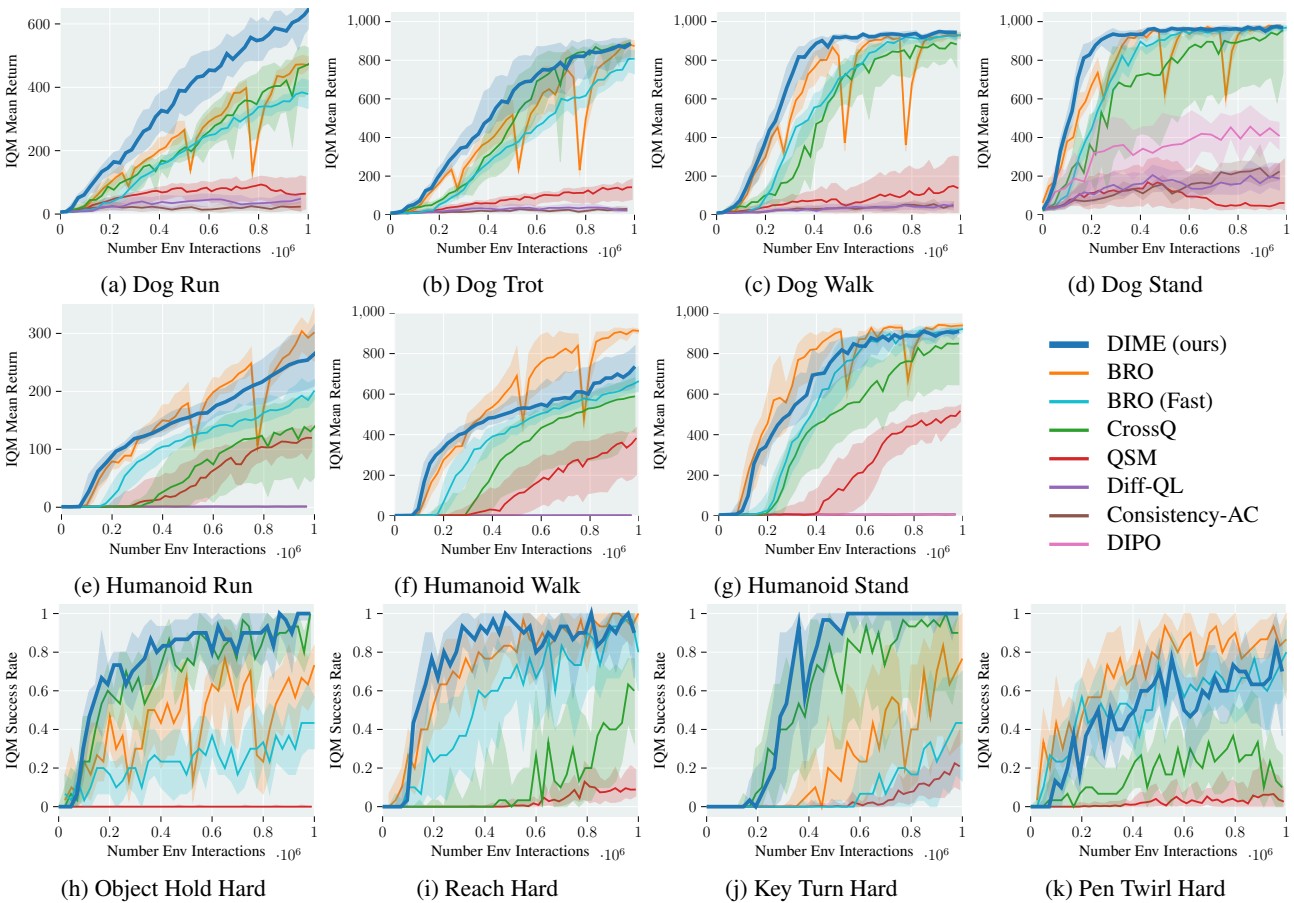

*Figure 4.* **Training curves on DMC's dog, humanoid tasks, and the hand environments from the MYO Suite.** DIME performs favorably on the high-dimensional dog tasks, where it significantly outperforms all baselines (dog-run) or converges faster to the final performance. On the humanoid tasks, DIME outperforms all diffusion-based baselines, CrossQ and BRO Fast, and performs on par with BRO on the humanoid-stand task and slightly worse on the humanoid-run and humanoid-walk tasks. In the MYO SUITE environments, DIME performs consistently on all tasks, either outperforming the baselines or performing on par.

mance for varying diffusion steps on DMC's humanoid-run environment and report the corresponding runtimes for 1 Mio environment steps in Fig. 3b on an *Nvidia A100* GPU machine. With an increasing number of diffusion steps, the performance and runtime increases. However, from 16 diffusion steps on, the performance stays the same.

### 5.2. Performance Comparisons

We consider environments with high-dimensional observation and action spaces from three benchmark suits for a robust performance assessment (please see Appendix C).

**Gym Environments.** Fig 3c and Fig. 3d show the learning curves for the *An-tv3* and *Humanoid-v3* tasks respectively. While the diffusion-based baselines perform reasonably well on the *Ant-v3* task with DIPO outperforming the rest, they are all outperformed by DIME and CrossQ which perform comparably. On the *Humanoid-v3* DIME achieves a significantly higher return than all baselines.

**DMC: Dog and Humanoid Tasks (Fig. 4).** We benchmark on DMC suit's challenging *dog* and *humanoid* environments, where we additionally consider BRO and BRO Fast as a Gaussian-based policy baseline. BRO Fast is identical to BRO but differs only in the update-to-data (UTD) ratio of two as DIME and CrossQ. Please note that we used the online available learning curves provided by the official implementation for BRO. DIME outperforms all baselines significantly on the *dog-run* environment and converges faster to the same end performance on the remaining dog environments (see Fig. 4a - 4d). BRO has slightly higher average performance on the *humanoid-run* and *humanoid-walk* (see Fig. 4f - 4e)) tasks indicating that DIME performs favorably on more high-dimensional tasks like the dog environments and tasks from the myo suite. However, DIME's asymptotic behavior in the *humanoid-run* achieves slightly higher aggregated performance than BRO, where we have run both algorithms for 3M steps (Fig. 6c). However, BRO requires full parameter resets leading to performance drops

during training and it is run with a UTD ratio of 10 which is 5 times higher than DIME. This leads to longer training times. As reported in their paper (Nauman et al., 2024), BRO needs an average training time of 8.5h, whereas DIME trains in approximately 4.5h with 16 diffusion steps on the humanoid-run with the same hardware (*Nvidia A100*).

**MYO Suite (Fig. 4).** Except for *pen twirl hard* (Fig. 4k), DIME consistently outperforms BRO and BRO Fast in that it converges to a higher or faster end success rate. DIME also consistently outperforms CrossQ in terms of the achieved success rates on all the tasks except for the object hold hard task 4h, where DIME converges faster.

## 6. Conclusion and Future Work

In this work, we introduced DIME, a method for learning diffusion models for maximum entropy reinforcement learning by leveraging connections to approximate inference. We view this work as a starting point for exciting future research. Specifically, we explored *denoising* diffusion models, where the forward process follows an Ornstein-Uhlenbeck process. However, approximate inference with diffusion models is an active and rapidly evolving field, with numerous recent advancements that consider alternative stochastic processes. For example, Richter & Berner proposed learning both the forward and backward processes, while Nusken et al. (2024) further enhanced exploration by incorporating the gradient of the target density into the diffusion process. Additionally, Chen et al. (2024) combined learned diffusion models with Sequential Monte Carlo (Del Moral et al., 2006), resulting in a highly effective inference method. These approaches hold significant promise for further improving diffusion-based policies in RL. We have conducted preliminary experiments on the framework from Richter & Berner and provide them in Appendix F. Finally, we note that the loss function used in this work (see Eq. 25) is based on the Kullback-Leibler divergence. However, in principle, any divergence could be used. For instance, the log-variance divergence (Richter & Berner) has shown promising results in optimizing diffusion models for approximate inference (Chen et al., 2024; Noble et al., 2024). Exploring alternative objectives could lead to additional performance improvements. Another interesting future research lies in investigating the effects of using more sophisticated critic structures, such as transformers, as proposed by Li et al. (2025).

## Acknowledgements

The authors acknowledge support from the state of Baden-Württemberg through the HoreKa supercomputer funded by the Ministry of Science, Research and the Arts Baden-Württemberg and by the German Federal Ministry of Education and Research. This work has been supported by the DFG Collaborative Research Center 1574, Circular Factory for the Perpetual Product, and by the pilot program Core Informatics of the Helmholtz Association (HGF). This research has been additionally supported by the DFG Emmy Noether project CH 2676/1-1. This research was also supported by the research cluster "Third Wave of AI", funded by the excellence program of the Hessian Ministry of Higher Education, Science, Research and the Arts, hessian.AI.

## Impact Statement

This paper presents work whose goal is to advance the field of Machine Learning. There are many potential societal consequences of our work, none of which we feel must be specifically highlighted here.

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

## A. Derivations

**Lower-Bound Derivation.** $\mathcal{H}(\pi_0(a_0|s)) \geq \ell_{\bar{\pi}}(a^0, s)$

$$\mathcal{H}(\pi_0(a_0|s)) = -\mathbb{E}_{\overleftarrow{\pi}_{0:N}}\left[\log \frac{\overleftarrow{\pi}_{0:N}(a^{0:N}|s)}{\overleftarrow{\pi}_{1:N|0}(a^{1:N}|s,a^0)}\right] \tag{29}$$

$$= -\mathbb{E}_{\overleftarrow{\pi}_{0:N}}\left[\log \frac{\overleftarrow{\pi}_{0:N}(a^{0:N}|s)\overrightarrow{\pi}_{1:N|0}(a^{1:N}|s,a^0)}{\overleftarrow{\pi}_{1:N|0}(a^{1:N}|s,a^0)\overrightarrow{\pi}_{1:N|0}(a^{1:N}|s,a^0)}\right]$$

$$= \mathbb{E}_{\overleftarrow{\pi}_{0:N}}\left[\log \frac{\overrightarrow{\pi}_{1:N|0}(a^{1:N}|s,a^0)}{\overleftarrow{\pi}_{0:N}(a^{0:N}|s)}\right] + \mathbb{E}_{\overleftarrow{\pi}_{0:N}}\left[\log \frac{\overleftarrow{\pi}_{1:N|0}(a^{1:N}|s,a^0)}{\overrightarrow{\pi}_{1:N|0}(a^{1:N}|s,a^0)}\right] \tag{30}$$

$$= \mathbb{E}_{\overleftarrow{\pi}_{0:N}}\left[\log \frac{\overrightarrow{\pi}_{1:N|0}(a^{1:N}|s,a^0)}{\overleftarrow{\pi}_{0:N}(a^{0:N}|s)}\right] + \mathbb{E}_{\pi_0}\left[D_{\mathrm{KL}}\left(\overleftarrow{\pi}_{1:N|0}(a^{1:N}|s,a^0)\| \overrightarrow{\pi}_{1:N|0}(a^{1:N}|s,a^0)\right)\right] \tag{31}$$

$$\geq \mathbb{E}_{\overleftarrow{\pi}_{0:N}}\left[\log \frac{\overrightarrow{\pi}_{1:N|0}(a^{1:N}|s,a^0)}{\overleftarrow{\pi}_{0:N}(a^{0:N}|s)}\right], \tag{32}$$

where we have used the relation

$$\pi_0(a_0|s) = \frac{\overleftarrow{\pi}_{0:N}(a^{0:N}|s)}{\overleftarrow{\pi}_{1:N|0}(a^{1:N}|s,a^0)} \tag{33}$$

and the fact that the KL divergence is always non-negative

**Approximate Inference Formulation.** Recall the definition of the Q-function

$$Q^{\overleftarrow{\pi}}(s_t, a_t^0) = r_t + \sum_{l=1} \gamma^l \mathbb{E}_{\rho_\pi}\left[r_{t+l} + \ell_{\bar{\pi}}(a_{t+l}^0, s_{t+l})\right]. \tag{34}$$

and

$$\ell_{\bar{\pi}}(a^0, s) = \mathbb{E}_{\overleftarrow{\pi}_{0:N}}\left[\log \frac{\overrightarrow{\pi}_{1:N|0}(a^{1:N}|a^0,s)}{\overleftarrow{\pi}_{0:N}(a^{0:N}|s)}\right]. \tag{35}$$

We start reformulating the objective

$$J(\overleftarrow{\pi}) \geq \bar{J}(\overleftarrow{\pi}) = \sum_{t=l}^{\infty} \gamma^{t-l}\mathbb{E}_{\rho_\pi}\left[r_t + \ell_{\bar{\pi}}(a_t^0, s_t)\right]. \tag{36}$$

$$= \sum_{t=l+1}^{\infty} \gamma^{t-l}\mathbb{E}_{\rho_\pi}\left[r_t + \ell_{\bar{\pi}}(a_t^0, s_t)\right] + \mathbb{E}_{\rho^\pi}\left[r_l + \ell_{\bar{\pi}}(a_l^0, s_l)\right] \tag{37}$$

$$= \mathbb{E}_{\rho^\pi}\left[Q^{\overleftarrow{\pi}}(s_t, a_t^0)\right] + \mathbb{E}_{\rho^\pi}\left[\ell_{\bar{\pi}}(a_l^0, s_l)\right] \tag{38}$$

$$= \mathbb{E}_{\rho^\pi}\left[Q^{\overleftarrow{\pi}}(s_t, a_t^0) + \ell_{\bar{\pi}}(a_l^0, s_l)\right] \tag{39}$$

$$= \mathbb{E}_{\rho^\pi, \overleftarrow{\pi}_{0:N}}\left[Q^{\overleftarrow{\pi}}(s_t, a_t^0) + \log \frac{\overrightarrow{\pi}_{1:N|0}(a^{1:N}|a^0,s)}{\overleftarrow{\pi}_{0:N}(a^{0:N}|s)}\right] \tag{40}$$

$$= -\mathbb{E}_{\rho^\pi}\left[D_{\mathrm{KL}}\left(\overleftarrow{\pi}(a^{0:N}|s)\| \overrightarrow{\pi}(a^{0:N}|s)\right) - \log \mathcal{Z}^{\overleftarrow{\pi}}(s)\right], \tag{41}$$

where we used

$$\overrightarrow{\pi}_0(a^0|s) = \frac{\exp Q^{\overleftarrow{\pi}}(s, a^0)}{\mathcal{Z}^{\overleftarrow{\pi}}(s)} \tag{42}$$

in the last step. When minimizing, the negative sign in front of the KL vanishes. Please note that the expectation over the marginal state distribution was ommited in the main text to avoid cluttered notation.

# B. Proofs

**Proof of Proposition 4.1 (Policy Evaluation).** Let's define the entropy-augmented reward of a diffusion policy as

$$r_{\bar{\pi}}(s_t, a_t^0) \triangleq r_t(s_t, a_t^0) + \mathbb{E}_{\bar{\pi}_{0:N}}\left[\log \frac{\vec{\pi}_{1:N|0}(a^{1:N}|a^0, s)}{\bar{\pi}_{0:N}(a^{0:N}|s)}\right] \tag{43}$$

and the update rule for the Q-function as

$$Q(s_t, a_t^0) \leftarrow r_{\bar{\pi}}(s_t, a_t^0) + \gamma \mathbb{E}_{s_{t+1} \sim p, a_{t+1}^0 \sim \bar{\pi}}\left[Q(s_{t+1}, a_{t+1}^0)\right]. \tag{44}$$

This formulation allows us to apply the standard convergence results for policy evaluation as stated in (Sutton & Barto, 1999).

**Proof of Proposition 4.2 (Policy Improvement).** It holds that

$$\bar{\pi}^{(i+1)}(a^{0:N}|s) = \frac{\exp Q^{\pi^{(i)}}(s, a^N)}{Z^{\pi^{(i)}}(s)} \vec{\pi}^{(i)}(a^{0:N-1}|a^N, s) \tag{45}$$

Moreover, using the fact that the KL divergence is always non-negative, we obtain

$$0 = D_{\text{KL}}\left(\bar{\pi}^{(i+1)}(a^{0:N}|s) \| \bar{\pi}^{(i+1)}(a^{0:N}|s)\right) \leq D_{\text{KL}}\left(\bar{\pi}^{(i)}(a^{0:N}|s) \| \bar{\pi}^{(i+1)}(a^{0:N}|s)\right) \tag{46}$$

Rewriting the KL divergences yields

$$\mathbb{E}_{\bar{\pi}^{(i+1)}}\left[\log \frac{\bar{\pi}^{(i+1)}(a^{0:N}|s)}{\bar{\pi}^{(i+1)}(a^{0:N}|s)}\right] \leq \mathbb{E}_{\bar{\pi}^{(i)}}\left[\log \frac{\bar{\pi}^{(i)}(a^{0:N}|s)}{\bar{\pi}^{(i+1)}(a^{0:N}|s)}\right] \tag{47}$$

$$\Longleftrightarrow \quad \mathbb{E}_{\bar{\pi}^{(i+1)}}\left[\log \bar{\pi}^{(i+1)}(a^{0:N}|s)\right] - \mathbb{E}_{\bar{\pi}^{(i+1)}}\left[\log \bar{\pi}^{(i+1)}(a^{0:N}|s)\right] \tag{48}$$
$$\leq \mathbb{E}_{\bar{\pi}^{(i)}}\left[\log \bar{\pi}^{(i)}(a^{0:N}|s)\right] - \mathbb{E}_{\bar{\pi}^{(i)}}\left[\log \bar{\pi}^{(i+1)}(a^{0:N}|s)\right]$$

$$\Longleftrightarrow \quad \mathbb{E}_{\bar{\pi}^{(i+1)}}\left[\log \bar{\pi}^{(i+1)}(a^{0:N}|s)\right] - \mathbb{E}_{\bar{\pi}^{(i+1)}}\left[\log \frac{\exp Q^{\pi^{(i)}}(s, a^N)}{Z^{\pi^{(i)}}(s)} \vec{\pi}^{(i)}(a^{0:N-1}|a^N, s)\right] \tag{49}$$
$$\leq \mathbb{E}_{\bar{\pi}^{(i)}}\left[\log \bar{\pi}^{(i)}(a^{0:N}|s)\right] - \mathbb{E}_{\bar{\pi}^{(i)}}\left[\log \frac{\exp Q^{\pi^{(i)}}(s, a^N)}{Z^{\pi^{(i)}}(s)} \vec{\pi}^{(i)}(a^{0:N-1}|a^N, s)\right]$$

$$\Longleftrightarrow \quad \mathbb{E}_{\bar{\pi}^{(i+1)}}\left[Q^{\pi^{(i)}}(s, a^N)\right] + \mathbb{E}_{\bar{\pi}^{(i+1)}}\left[\log \frac{\vec{\pi}^{(i)}(a^{0:N-1}|a^N, s)}{\bar{\pi}^{(i+1)}(a^{0:N}|s)}\right] \tag{50}$$
$$\geq \mathbb{E}_{\bar{\pi}^{(i)}}\left[Q^{\pi^{(i)}}(s, a^N)\right] + \mathbb{E}_{\bar{\pi}^{(i)}}\left[\log \frac{\vec{\pi}^{(i)}(a^{0:N-1}|a^N, s)}{\bar{\pi}^{(i)}(a^{0:N}|s)}\right]$$

To keep the notation uncluttered we use

$$d^{(i+1)}(s, a^N) = \mathbb{E}_{\bar{\pi}^{(i+1)}}\left[\log \frac{\vec{\pi}^{(i)}(a^{0:N-1}|a^N, s)}{\bar{\pi}^{(i+1)}(a^{0:N}|s)}\right] \quad \text{and} \quad d^{(i)}(s, a^N) = \mathbb{E}_{\bar{\pi}^{(i)}}\left[\log \frac{\vec{\pi}^{(i)}(a^{0:N-1}|a^N, s)}{\bar{\pi}^{(i)}(a^{0:N}|s)}\right] \tag{51}$$

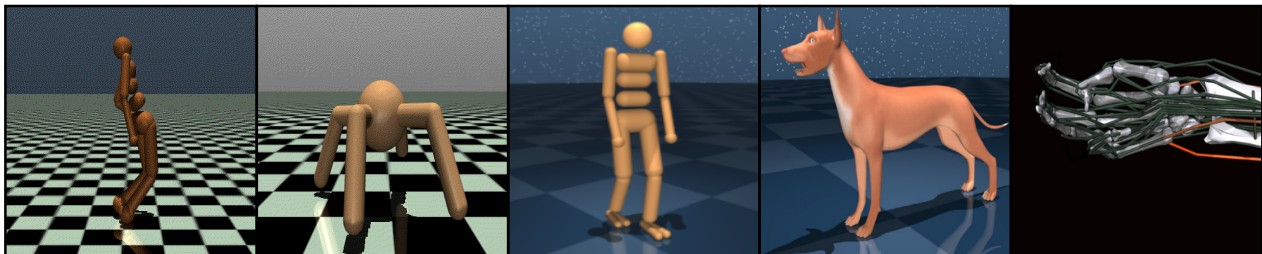

*Figure 5.* **Considered environments.** The *Humanoid-v3* and the *Ant-v3* are environments from the mujoco gym benchmark (Brockman et al., 2016). The three environments *humanoid-run*, *humanoid-walk* and *humanoid-stand* are from the deepmind control suite (DMC) benchmark (Tunyasuvunakool et al., 2020). The dog environments consist of *dog-run*, *dog-walk*, *dog-stand*, *dog-trot* and are also from the DMC sutie benchmark. Finally, the myo suite hand environments *object-hold-hard*, *reach-hard*, *key-turn-hard*, *pen-twirl-hard* are from the myo suite (Caggiano et al., 2022).

$$Q^{\pi^{(i)}}(s, a^N) = r_0 + \mathbb{E}\left[\gamma\left(d^{(i)}(s_1, a_1^N) + \mathbb{E}_{\bar{\pi}^{(i)}}\left[Q^{\pi^{(i)}}(s_1, a_1^N)\right]\right)\right] \tag{52}$$

$$\leq r_0 + \mathbb{E}\left[\gamma\left(d^{(i+1)}(s_1, a_1^N) + \mathbb{E}_{\bar{\pi}^{(i+1)}}\left[Q^{\pi^{(i)}}(s_1, a_1^N)\right]\right)\right] \tag{53}$$

$$= r_0 + \mathbb{E}\left[\gamma\left(d^{(i+1)}(s_1, a_1^N) + r_1\right) + \gamma^2\left(d^{(i)}(s_2, a_2^N) + \mathbb{E}_{\bar{\pi}^{(i)}}\left[Q^{\pi^{(i)}}(s_2, a_2^N)\right]\right)\right] \tag{54}$$

$$\leq r_0 + \mathbb{E}\left[\gamma\left(d^{(i+1)}(s_1, a_1^N) + r_1\right) + \gamma^2\left(d^{(i+1)}(s_2, a_2^N) + \mathbb{E}_{\bar{\pi}^{(i+1)}}\left[Q^{\pi^{(i)}}(s_2, a_2^N)\right]\right)\right] \tag{55}$$

$$\vdots \tag{56}$$

$$\leq r_0 + \mathbb{E}\left[\sum_{t=1}^{\infty}\gamma^t\left(d^{(i+1)}(s_t, a_t^N) + r_t\right)\right] = Q^{\pi^{(i+1)}}(s, a^N) \tag{57}$$

Since $Q$ improves monotonically, we eventually reach a fixed point $Q^{(i+1)} = Q^{(i)} = Q^*$

**Proof of Proposition 4.3 (Policy Iteration).** From Proposition 4.2 it follows that $Q^{\bar{\pi}^{i+1}}(s, a) \geq Q^{\bar{\pi}^i}(s, a)$. If for $\lim_{k\to\infty}\bar{\pi}^k = \bar{\pi}^*$, then it must hold that $Q^{\bar{\pi}^*(s,a)} \geq Q^{\bar{\pi}}(s, a)$ for all $\bar{\pi} \in \bar{\Pi}$ which is guaranteed by Proposition 4.2.

## C. Environments

All environments are visualized in Fig. 5. We consider the *Ant-v3* and the *Humanoid-v3* environments from mujoco gym (Brockman et al., 2016). The *humanoid-stand*, *humanoid-walk* , *humanoid-run*, *dog-stand*, *dog-walk*, *dog-trot* and *dog-run* environments from the deepmind control suite (DMC) (Tunyasuvunakool et al., 2020). The hand environments from myo suite are the *object-hold-random*, *reach-random*, *key-turn-random* and *pen-twirl-random* environments (Caggiano et al., 2022). The action and observation spaces of the respective environments are shown in Table 1.

## D. Implementation Details

We consider a score network with *3* layers and a *256* dimensional hidden layer with gelu activation function. We use Fourier features to encode the timestep and scale the embedding using a feed-forward neural network with two layers, with a hidden dimension of 256. For the diffusion coefficient, we use a cosine schedule and additionally optimize a scaling parameter for the diffusion coefficient per dimension end-to-end (i.e,. we learn the parameter $\beta$ (please see Appendix G).

We employ distributional Q following (Bellemare et al., 2017), where the Q-model outputs probabilities $q$ over $b$ bins. Using the bellman backup operator for diffusion models from Eq. 21 and the bin values $b$ we follow (Bellemare et al., 2017) and calculate the target probabilities $q_{target}$. Using the entropy-regularized cross-entropy loss $\mathcal{L}(\phi) = -\sum q_{target}\log q_\phi - 0.005\sum q_\phi \log q_\phi$ we update the parameters $\phi$ of the Q-function. Please note that the entropy regularization was not proposed in the original paper from (Bellemare et al., 2017), however, we noticed that a small regularization helps improve

| Training Environment | Observation Space Dim. | Action Space Dim. |
|---|---|---|
| Ant-v3 | 111 | 8 |
| Humanoid-v3 | 376 | 17 |
| dog-run | 223 | 38 |
| dog-walk | 223 | 38 |
| dog-trot | 223 | 38 |
| dog-stand | 223 | 38 |
| humanoid-run | 67 | 24 |
| humanoid-walk | 67 | 24 |
| humanoid-stand | 67 | 24 |
| myoHandObjHoldRandom-v0 | 91 | 39 |
| myoHandReachRandom-v0 | 115 | 39 |
| myoHandKeyTurnRandom-v0 | 93 | 39 |
| myoHandPenTwirlRandom-v0 | 83 | 39 |

*Table 1.* Observation and Action Space Dimensions for Various Training Environments

the performance in the early learning stages but does not change the asymptotic performance. Additionally, we follow (Nauman et al., 2024) and use the *mean* of the two Q-values instead of the *min* as it has usually been used in RL so far.

The expected Q-values for updating the actor can be easily calculated using the expectation $Q(s, a_t^0) = \sum_i q_i(s_t, a_t^0) b_i$

**Action Scaling.** Practical applications have a bounded action space that can usually be scaled to a fixed range. However, the action range of the diffusion policy $\bar{\pi}$ is unbounded. Therefore, we follow recent works (Haarnoja et al., 2018b) and propose applying the change of variables with a *tanh* squashing function at the last diffusion step $n = 0$. For the backward process $\bar{q}_{0:N}(u^{0:N}|s)$ with unbounded action space $u \in \mathbb{R}^D$ we can squash the action $a^0 = \tanh u^0$ such that $a^0 \in (-1, 1)$ and its density is given by

$$\bar{\pi}_{0:N}(a^{0:N}|s) = \bar{q}_{0:N}(u^{0:N}|s) \det \left| \left( \frac{\mathrm{d}a^0}{\mathrm{d}u^0} \right) \right|^{-1}, \tag{58}$$

with the corresponding log-likelihood

$$\log \bar{\pi}_{0:N}(a^{0:N}|s) = \log \bar{q}_N(u^N) + \sum_{n=1}^{N} \log \bar{q}_{n-1}(u^{n-1}|u^n, s) - \sum_{i=1}^{D} \log \left( 1 - \tanh^2 \left( u_i^N \right) \right). \tag{59}$$

This means that the Gaussian kernels of the diffusion chain have the same log probabilities except for the correction term of the last step at $n = 0$.

---

**Algorithm 1** DIME: Diffusion-Based Maximum Entropy Reinforcement Learning

---

**Input:** Initialized parameters $\theta, \phi, \alpha$, learningrates $\lambda$
1: **for** $k = 1$ to M **do**
2:    **if** k % UTD **then**
3:       $a_t^{0:T} \sim \pi_{0:N}^\theta(a^{0:N}|s_t)$
4:       $s_{t+1} \sim p(s_{t+1}|a_t^0, s_t)$
5:       $\mathcal{D} \leftarrow \mathcal{D} \bigcup \{s_t, a_t^0, r_t, s_{t+1}\}$
6:    **end if**
7:    $\phi \leftarrow \phi - \lambda_\phi \nabla_\phi J_Q(\phi)$ (Eq. 24)
8:    **if** k % POLICYDELAY **then**
9:       $\theta \leftarrow \theta - \lambda_\theta \nabla_\theta \mathcal{L}(\theta)$ (Eq. 25)
10:      $\alpha \leftarrow \alpha - \lambda_\alpha J(\alpha)$ (Eq. 28)
11:    **end if**
12: **end for**

---

Algorithm 1 shows the learning procedure of DIME. Note that policy delay refers to the number of delayed updates of the policy compared to the critic. UTD is the update to data ratio.

## E. List of Hyperparameters

Please note that we have used the official code releases of the respective baseline methods for training. For BRO and BRO Fast we used the provided learning curves

| | DIME | QSM | Diff-QL | Consistency-AC | DIPO | DACER | QVPO |
|---|---|---|---|---|---|---|---|
| Update-to-data ratio | 2 | 1 | 1 | 1 | 1 | 1 | 1 |
| Discount | 0.99 | 0.99 | 0.99 | 0.99 | 0.99 | 0.99 | 0.99 |
| batch size | 256 | 256 | 256 | 256 | 256 | 256 | 256 |
| Buffer size | 1e6 | 1e6 | 1e5 | 1e5 | 1e6 | 1e6 | 1e6 |
| $\mathcal{H}_{target}$ | 4dim($\mathcal{A}$) | N/A | N/A | N/A | N/A | -0.9dim$\mathcal{A}$ | N/A |
| Critic hidden depth | 2 | 2 | 2 | 3 | 3 | 3 | 2 |
| Critic hidden size | 2048 | 2048 | 256 | 256 | 256 | 256 | 256 |
| Actor/Score depth | 3 | 3 | 4 | 4 | 4 | 3 | 2 |
| Actor/Score size | 256 | 256 | 256 | 256 | 256 | 256 | 256 |
| Num. Bins/Quantiles | 100 | N/A | N/A | N/A | N/A | 2 | N/A |
| Temp. Learn. Rate | 1e-3 | N/A | N/A | N/A | N/A | 3e-2 | N/A |
| Learn. Rate Critic | 3e-4 | 3e-4 | 3e-4 | 3e-4 | 3e-4 | 3e-4 | 3e-4 |
| Learn. Rate Actor/Score | 3e-4 | 3e-4 | 1e-5 | 1e-5 | 3e-4 | 3e-4 | 3e-4 |
| Optimizer | Adam | Adam | Adam | Adam | Adam | Adam | Adam |
| Diffusion Steps | 16 | 15 | 5 | N/A | 100 | 20 | 20 |
| Prior Distr. | $\mathcal{N}(0, 2.5)$ | $\mathcal{N}(0, 1)$ | N/A | N/A | N/A | $\mathcal{N}(0, 1)$ | $\mathcal{N}(0, 1)$ |
| Exploration Steps | 5000 | 1e4 | 1e4 | 1e4 | 1e4 | 1e4 | 1e4 |
| Score-Q align. factor | N/A | 50 | N/A | N/A | N/A | N/A | N/A |

*Table 2.* Hyperparameters of DIME and all diffusion-based algorithms for all benchmark suits. Varying hyperparameters for different benchmark suits are described in the text.

| | DIME | BRO | BRO Fast | CrossQ |
|---|---|---|---|---|
| Polyak weight | N/A | 0.005 | 0.005 | N/A |
| Update-to-data ratio | 2 | 10 | 2 | 2 |
| Discount | 0.99 | 0.99 | 0.99 | 0.99 |
| batch size | 256 | 128 | 128 | 256 |
| Buffer size | 1e6 | 1e6 | 1e6 | 1e6 |
| $\mathcal{H}_{target}$ | 4dim($\mathcal{A}$) | dim($\mathcal{A}$)/2 | dim($\mathcal{A}$)/2 | dim($\mathcal{A}$) |
| Critic hidden depth | 2 | BRONET | BRONET | 2 |
| Critic hidden size | 2048 | 512 | 512 | 2048 |
| Actor/Score depth | 3 | BRONET | BRONET | 3 |
| Actor/Score size | 256 | 256 | 256 | 256 |
| Num. Bins/Quantiles | 100 | 100 | 100 | N/A |
| Temp. Learn. Rate | 1e-3 | 3e-4 | 3e-4 | 3e-4 |
| Learn. Rate Critic | 3e-4 | 3e-4 | 3e-4 | 7e-4 |
| Learn. Rate Actor/Score | 3e-4 | 3e-4 | 3e-4 | 7e-4 |
| Optimizer | Adam | AdamW | AdamW | Adam |
| Diffusion Steps | 16 | N/A | N/A | N/A |
| Prior Distr. | $\mathcal{N}(0, 2.5)$ | N/A | N/A | N/A |
| Exploration Steps | 5000 | 2500 | 2500 | 5000 |

*Table 3.* Hyperparameters of DIME and Gaussian-based algorithms for all benchmark suits. Varying hyperparameters for different benchmark suits are described in the text.

**DIME.** For DIME, we use distributional Q, where the maximum and minimum values for the bins have been chosen per benchmark suite. We have used $v_{min} = -1600$ and $v_{max} = 1600$ for the gym environments, $v_{min} = -200$ and $v_{max} = 200$ for the DMC suite and $v_{min} = -3600$ and $v_{max} = 3600$ for the myo suite.

**QSM**. In certain environments, we observed that QSM with default hyperparameters performed poorly, particularly in several DMC tasks and the Gym Ant-v3 tasks. To address this, we fine-tuned the hyperparameters for QSM in each of these underperforming tasks. For the DMC tasks, we found that QSM often requires an $\alpha$ value—representing the alignment factor between the score and the Q-function (Psenka et al., 2024)—in the range of 100-200, rather than the default value of 50 reported in QSM's original implementation. In the Ant-v3 task, we determined that $\alpha$ needs to be set to 1. In the original implementation, the number of diffusion steps is set to be 5, however, we found using more steps, such as 10 and 15, can significantly improve the performance in these under performed tasks.

**CrossQ.** We used the hyperparameters from the original paper (Bhatt et al., 2024) for the gym benchmark suite. However, we used a different set of hyperparameters for the DMC and MYO suites for improved performance. More precisely, we increased the policy size to *3 layers* with *256 hidden size*. Additionally, we reduced the learning rate to *7e-4*.

## F. General Diffusion Policies

DIME's maximum entropy reinforcement learning framework for training diffusion policies is not specifically restricted to denoising diffusion policies but can be extended to general diffusion policies. This can be realized using the General Bridges framework as presented in (Richter & Berner). In this case, we can write the forward and backward process as

$$\mathrm{d}a_t = [f(a_t, t) + \beta u(a_t, s, t)] \, \mathrm{d}t + \sqrt{2\beta_t} \mathrm{d}B_t, \qquad a_0 \sim \vec{\pi}_0(\cdot|s), \tag{60}$$

$$\mathrm{d}a_t = [f(a_t, t) - \beta v(a_t, s, t)] \, \mathrm{d}t + \sqrt{2\beta_t} \mathrm{d}B_t, \qquad a_T \sim \mathcal{N}(0, I), \tag{61}$$

with the drift and control functions $f, u, v : \mathbb{R}^d \times [0, T] \to \mathbb{R}^d$, the diffusion coefficient $\beta : [0, T] \to \mathbb{R}^+$, standard Brownian motion $(B_t)_{t \in [0,T]}$ and some target policy $\vec{\pi}_0$. Again we denote the marginal density of the forward process as $\vec{\pi}_t$ and the conditional density at time $t$ given $l$ as $\vec{\pi}_{t|l}$ for $t, l \in [0, T]$. The backward process starts from $\overleftarrow{\pi}_T = \vec{\pi}_T \sim \mathcal{N}(0, I)$ and runs backward in time where we denote its density as $\overleftarrow{\pi}$.

The respective discretization using the Euler Maruyama (EM) (Särkkä & Solin, 2019) method are given by

$$a^{n+1} = a^n + [f(a^n, n) + \beta u(a^n, s, n)] \, \delta + \epsilon_n, \tag{62}$$

$$a^{n-1} = a^n - [f(a^n, n) - \beta v(a^n, s, n)] \, \delta + \xi_n, \tag{63}$$

where $\epsilon_n, \xi_n \sim \mathcal{N}(0, 2\beta\delta I)$, with the constant discretization step size $\delta$ such that $N = T/\delta$ is an integer. We have used the simplified notation where we write $a^n$ instead of $a^{n\delta}$. The discretizations admit the joint distributions

$$\vec{\pi}_{0:N}(a^{0:N}|s) = \pi_0(a^0|s) \prod_{n=0}^{N-1} \vec{\pi}_{n+1|n}(a^{n+1}|a^n, s), \tag{64}$$

$$\overleftarrow{\pi}_{0:N}(a^{0:N}|s) = \overleftarrow{\pi}_N(a^N|s) \prod_{n=1}^{N} \overleftarrow{\pi}_{n-1|n}(a^{n-1}|a^n, s), \tag{65}$$

with Gaussian kernels

$$\vec{\pi}_{n+1|n}(a^{n+1}|a^n, s) = \mathcal{N}(a^{n+1}|a^n + [f(a^n, n) + \beta u(a^n, s, n)] \, \delta, 2\beta\delta I) \tag{66}$$

$$\overleftarrow{\pi}_{n-1|n}(a^{n-1}|a^n, s) = \mathcal{N}(a^{n-1}|a^n - [f(a^n, n) - \beta v(a^n, s, n)] \, \delta, 2\beta\delta I) \tag{67}$$

Following the same framework presented in the main text, we can now optimize the controls $u$ and $v$ using the same objective

$$\bar{\mathcal{L}}(u, v) = D_{\mathrm{KL}} \left( \overleftarrow{\pi}_{0:N}(a^{0:N}|s) | \vec{\pi}_{0:N}(a^{0:N}|s) \right), \tag{68}$$

where the target policy at time step $n = 0$ is given as

$$\pi_0(a^0|s) = \frac{\exp Q^{\overleftarrow{\pi}}(s, a^0)}{\mathcal{Z}^{\overleftarrow{\pi}}(s)}. \tag{69}$$

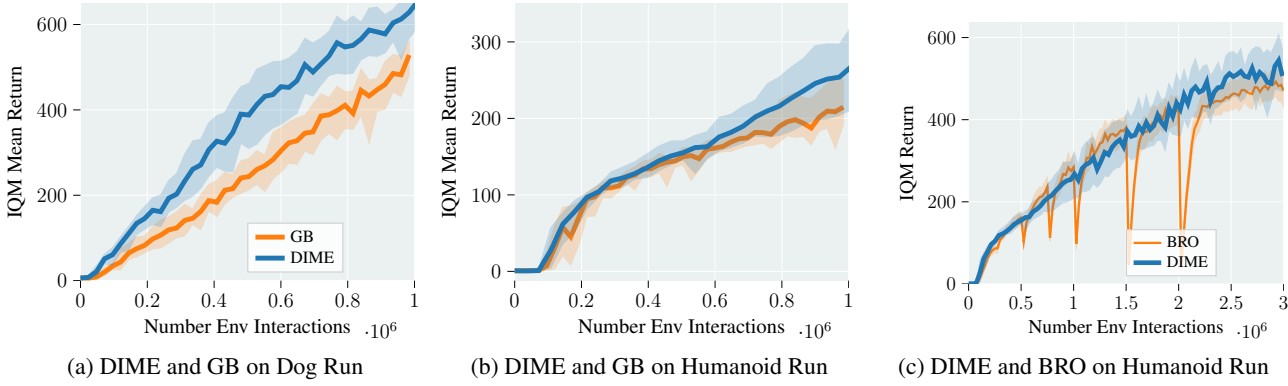

*Figure 6.* **Preliminary results for the GB sampler on the dog run (a) and humanoid run (b) environments from DMC. Comparison to BRO on the humanoid run for 3 million steps.**

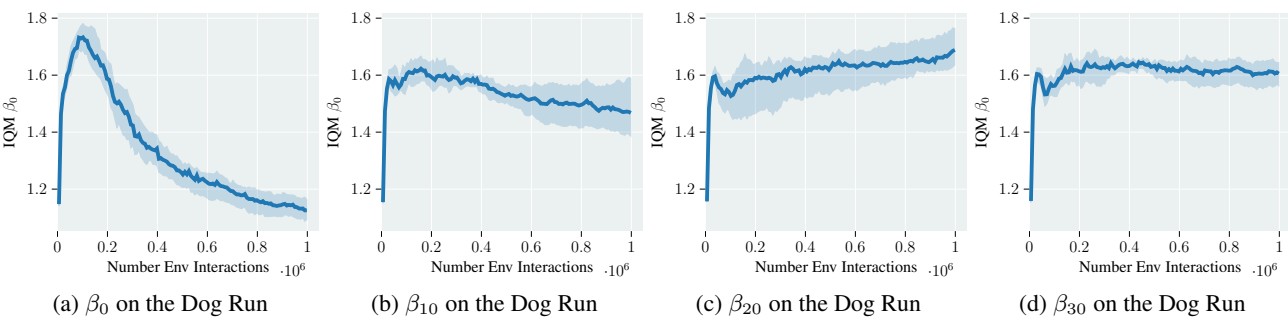

*Figure 7.* **Learned $\beta$ parameters.** DIME's policy improvement objective (Eq. 27) allows to train various parameters end-to-end, such as the scaling for the diffusion coefficient $\beta$. More concretely, we train a scaling parameter $\beta_k$ per dimension $k$, that scales the cosine schedule. We visualize the adaptation of the parameter for the dimension $k = 0, 10, 20, 30$ over the training, averaged over 10 seeds for the dog-run task. Clearly, DIME first increases the parameter at the beginning of the training phase. Depending on the dimension, it either converges to a rather high value ($k = 20$ and $k = 30$), or keeps being reduced for other dimensions $k = 0$ and $k = 10$.

In practice, we optimize the control functions $u$ and $v$ using parameterized neural networks. We have run preliminary results using the general bridge framework within the maximum entropy objective as suggested in our work. The learning curves can be seen in Fig. 6.

## G. Additional Experiments

**End-To-End Learning of $\beta$.** We visualize the adaptation of the scaling for the diffusion coefficient $\beta$ in Fig. 7 during learning on DMC's dog-run environment.

**Extended Analysis on Distributional Q Learning.** DIME employs distributional Q Learning (Bellemare et al., 2017) to represent the Q-function as a distribution over bins. We compare DIME to baselines when using distributional Q Learning and when using the well-known Bellman residual (see Eq. 24) for updating the parameters of the Q-function.

We start by comparing DIME with distributional Q learning against diffusion-based baselines that employ distributional Q learning. Fig. 8a and Fig. 8b show the learning curves on the *Ant-v3* and *Humanoid-v3*, respectively, where we compare against DACER, a distributional Q variant of Diff-QL, and Consistency-AC. DIME converges faster to the same performance as DACER on the *Ant-v3* task and outperforms the baselines on the *Humanoid-v3* task. In the setting without distributional Q Learning, i.e., when updating the parameters using the residual Bellman function, DIME performs similarly to DIPO and QVPO on the *Ant-v3* task and outperforms all baselines on the higher-dimensional *Humanoid-v3* task (Fig. 8c and Fig. 8d).

Additionally, we compare DIME with and without distributional Q Learning on the four dog environments from the DMC suite (Fig. 8), where we concentrate on the strong baselines BRO (Nauman et al., 2024) and CrossQ (Bhatt et al., 2024). BRO employs quantile distributional Q learning, whereas CrossQ uses the Bellman residual loss function for updating the

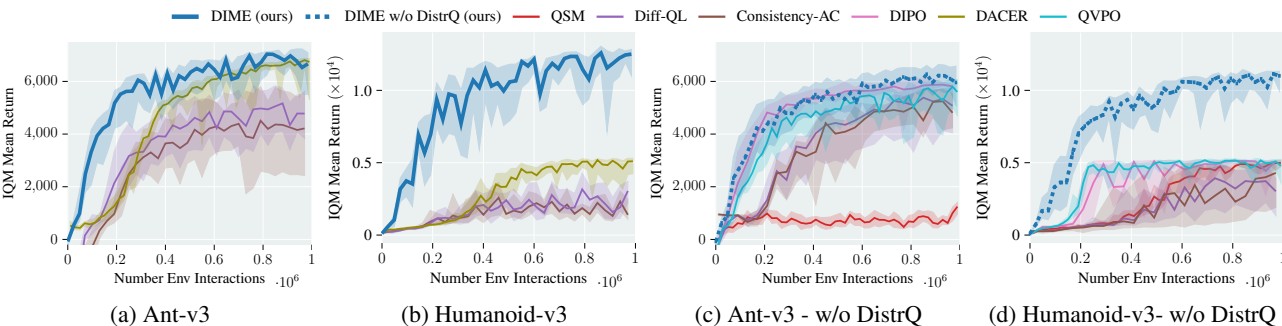

Figure 8. **Comparison to Diffusion Baselines with (a)-b)) and without Distributional Q (c)-d)) on the Ant-v3 and Humanoid-v3 tasks.** We provide the learning curves for distributional versions for Diff-QL and Consistency-AC alongside DACER, which employs distributional Q by default on the Ant-v3 (a) and Humanoid-v3 (b) tasks. DIME converges faster on the Ant-v3 (a) task to the same performance achieved by DACER and outperforms all baselines on the more high-dimensional Humanoid-v3 (b) task. Additionally, we compare DIME without distributional Q against the diffusion baselines without distributional Q on the Ant-v3 (c) and Humanoid-v3 (d) tasks. DIME without distributional Q performs on par with the baselines DIPO and QVPO on the Ant-v3 (c) and outperforms all baselines on the Humanoid-v3 (d).

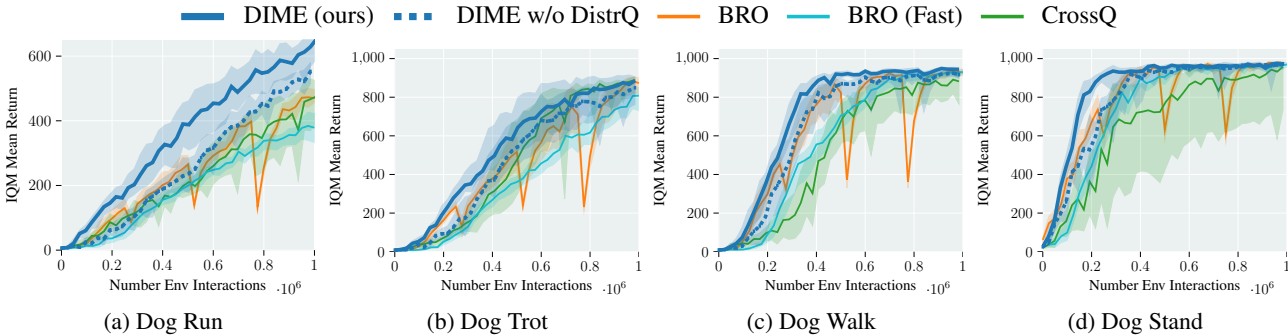

Figure 9. **Ablation on Distributional Q.** Comparison of DIME and DIME without employing distributional Q (dashed line). While there is a small improvement when using distributional Q, DIME w/o Distributional Q still performs on par, or better than BRO, which employs quantile distributional RL. DIME w/o DistrQ outperforms CrossQ and BRO (Fast).

Q-function's parameters. In the main text, we have already observed that DIME with distributional Q performs favorably over the baselines. Fig. 8 shows a small improvement when using distributional Q. However, DIME without distributional Q (dashed line) still performs on par, or better than BRO and consistently performs better than BRO (Fast) and CrossQ. Please note that BRO and BRO (Fast) employ quantile distributional RL (Nauman et al., 2024).

