# OpenReview forum: "DIME: Diffusion-Based Maximum Entropy Reinforcement Learning"
_ICML.cc/2025/Conference — ICML 2025 poster_

### Official Review · Reviewer_gx83 · 2025-03-08

**Overall Recommendation:** 4

**Summary:**

This paper proposed a diffusion-based maximum entropy RL. The authors first derived the lower bound of entropy by data processing inequality. Then the soft policy iteration is recased to diffusion-based version with new regularization term with the lower bound. The policy optmization is reformulated to the lower bound of soft policy iteration, which gives a tractble loss function to optimize diffusion policy. Experimental results have shown the performance is improved compared to previous diffusion-based RL.

**Claims And Evidence:**

1. The lower bound claim is correct.
2. The policy iteration scheme is correct.
3. The claim
> significantly outperforming other diffusion-based methods on challenging high-dimensional control benchmarks
is not that well supported, given that the authors only compare DIME (proposed) with QSM and DIPO, leaving DACER, QVPO, and other related works mentioned in the paper not involved.

**Essential References Not Discussed:**

No, the authors provides sufficient references.

**Experimental Designs Or Analyses:**

There is one thing not clear: the authors made two modifications to the current maximum entropy RL, one is the lower-bound modified soft policy interation, another is the Q-learning part via crossQ. The authors does not provide ablation studies that ablates either of these two modifications. How they contributes to the performance improvement is not clear.

**Update after rebuttal**: What I mean is the effect of crossQ, and the authors have resolved it in the rebuttal.

**Methods And Evaluation Criteria:**

1. The evaluation metrics make sense.
2. There is one thing not clear: the authors made two modifications to the current maximum entropy RL, one is the lower-bound modified soft policy interation, another is the Q-learning via crossQ. The authors does not provide ablation studies that ablates either of these two modifications. How they contributes to the performance improvement is not clear.

**Other Comments Or Suggestions:**

In line 351, There is a typo on the env names An-tv3, it should be Ant-v3.

**Other Strengths And Weaknesses:**

Strength:
1. The baselines keep up with some most recent RL algorithms, which is good.

Weakness:
1. The notations are extremely comfusing and have some errors. For example, I believe $\rightarrow$ means forward diffusion process and $\leftarrow$ means reverse diffusion process. However, in (27), you write $\pi^\rightarrow$ with $n-1|n$, which is not consistent since $n-1|n$ should be the reverse process Gaussian. I didn't fully understand.
2. As I said the policy loss stands on itself, the value of all the previous discussion remain unclear.

**Questions For Authors:**

I will repeat some questions I listed before,

---
1. I have concerns on the equation (25), which is the policy loss. It can stands on it self since it just fits the reverse diffusion trajectories with the forward diffusion trajectories. Then why all the previous lower bounds are necessary? Can we just do normal $Q$-learning and optimize (25)?
---
2. The claim near line 295
> .... stochastic opitmization of L(θ) does not need access to samples a0 ∼ exp Qϕ /Zϕ , instead relying
on stochastic gradients obtained via reparameterization trick
(Kingma, 2013) using samples from the diffusion model π θ .

seems problematic. In (27), no terms need reparametrization since it only involves the log probability. How exactly the reparametrization works here?

---
3. Continuing on 3, I check the code the authors provided, it shows the policy loss as
```python
actor_loss = (- min_qf_pi + ent_coef_value * (run_costs.squeeze() + sto_costs.squeeze() + terminal_costs.squeeze())).mean()
```
where the first term is $Q_\phi(s, \tilde a)$ where $\tilde a$ is sampled from $\pi^\theta$. However, in (27) the $a^0$ in $Q_\phi(s,a^0)$ term should not be a function of $\theta$ since it is derived from a part of $\pi^{\rightarrow}(a^{0:N}|s)$. Therefore, the implementation is actually mismatched with the actual policy loss in (27). Please kindly explain the gap between theoritical results and your implentation and how it affects the contribution of this paper.

**Update after rebuttal**: it is a misunderstanding from my side. The derivation is clear.

Overall, I do think there are some insights on the loss function in (25), which somehow flipped the KL divergence to make the diffusion policy training tractable. However, as the importance of major theoritical results remain unclear and the implementation are not matching the theoritical derivation and statements, I lean to reject the paper unless the authors have covincing answers to all these questions.

**Relation To Broader Scientific Literature:**

There might be some but none is worth mentioning here.

**Theoretical Claims:**

1. The lower bound and policy iteration convergence is correct.
---
2. I have concerns on the equation (25), which is the policy loss. It can stands on it self since it just fits the reverse diffusion trajectories with the forward diffusion trajectories. Then why all the previous lower bounds are necessary? Can we just do normal $Q$-learning and optimize (25)?
---
3. The claim near line 295
> .... stochastic opitmization of L(θ) does not need access to samples a0 ∼ exp Qϕ /Zϕ , instead relying
on stochastic gradients obtained via reparameterization trick
(Kingma, 2013) using samples from the diffusion model π θ .

seems problematic. In (27), no terms need reparametrization since it only involves the log probability.

---

> ### Author Rebuttal · Authors · 2025-03-31
>
> We thank the reviewer for taking the time to review our work and for the many helpful comments and suggestions. We hope the following replies address the questions and concerns raised.
>
> ---
>
> > The claim 'significantly outperforming other diffusion-based methods on challenging high-dimensional control benchmarks' is not that well supported, given that the authors only compare DIME (proposed) with QSM and DIPO, leaving DACER, QVPO […]
>
> We have conducted additional experiments and compared DIME to QVPO and DACER. We kindly refer the reviewer to our answers to Reviewer TJZx who shared a similar concern.
>
> ---
>
> > […] the authors made two modifications to the current maximum entropy RL , one is the lower-bound modified soft policy interation, another is the Q-learning via crossQ. The authors does not provide ablation studies that ablates either of these two modifications. How they contributes to the performance improvement is not clear.
>
> Please note that a direct comparison to the common max. entr. objective is not possible, as the marginal entropy of the diffusion process is not tractable.  Therefore, DIME proposes a lower-bound objective that is tractable and can be used to optimize the diffusion policy by adapting the policy iteration scheme. Please note that we only use CrossQ’s Q-network architecture which does not entail a modification to the maximum entropy objective. Our work proposes a framework for training diffusion policies that is orthogonal to improvements for critics/Q-functions.
>
> ---
>
> > I have concerns on the equation (25) which is the policy loss. [….] Then why all the previous lower bounds are necessary? Can we just do normal Q-learning and optimize (25)?
>
> Please note that optimizing the maximum entropy objective for diffusion models is problematic as the marginal entropy of the diffusion process is not tractable and requires further treatment. As such, our work proposes (25), which is justified through the lower bound in (16) or the data processing inequality in (20). Both perspectives show that optimizing (25) is equivalent to maximizing the maximum entropy objective in (1).
>
> We kindly ask the reviewer to elaborate on what is meant by 'normal Q-learning'. If the reviewer means defining the Q-function as the sum of the discounted rewards without taking the expected entropies into account, we would like to mention that this objective is equivalent to entropy-regularized RL [1, 2] and does not optimize the maximum entropy objective as we consider in this work.  We hope that this addresses the reviewers’ concerns. If we misunderstand the question, we are more than happy to provide further clarification.
>
> [1] Neu, G. A unified view of entropy-regularized markov decision processes.
>
> [2] Levine, S. Reinforcement Learning and Control as Probabilistic Inference.
>
> ---
>
>
> > […] However, in (27), you write  $\overrightarrow{\pi}$  with $n-1|n$, which is not consistent since $n-1|n$ should be the reverse process Gaussian.
>
> We thank the reviewer for spotting the mistake in (27) and apologize for any confusion it may have caused.  The correct transition densities read $\pi^{\theta}_{n-1|n}$ for the denoising process and $\overrightarrow{\pi} _{n|n-1}$ for the noising process.
>
> ---
>
> > The claim near line 295 […] In (27), no terms need reparameterization since it only involves the log probability. How exactly the reparameterization wokrs here? […] The implementation is actually mismatched with the actual policy loss in (27).
>
> Please note that the reparameterization trick is necessary as we are computing gradients with respect to the parameters of the distribution that we are taking expectations with. For $L(\theta$) (Eq. 27) he expectation is taken w.r.t. the parameterized denoising process $\pi^{\theta}\_{0:N}$.
> For diffusion models, we need to iteratively apply the reparameterization trick for every transition density $\pi^{\theta}\_{n-1|n}$  in order to reparameterize the joint distribution $\pi\_{0:N}^{\theta}$. For a single transition $\pi^{\theta}\_{n-1|n}$ the reparameterization trick reads
> $$
> a^{n-1}  = a^{n} + (\beta_{n} a^{n} + 2\eta^2\beta_{n} f_n^{\theta}(a^{n}|s))\delta + \xi_{n},
> $$
>
> with $\xi\_n \sim \mathcal{N}(0,2\eta^2\beta\_{n}\delta I)$, which agrees with the Euler-Maruyama discretization.  As such, the denoised action $a^0$ is a function of all previous transitions and therefore depends on $\theta$.  This is inline with (27) where the expectation is taken under the diffusion policy $\pi^{\theta}\_{0:N}$ which agrees with the implementation in our code base, where $a^0$ is coming from $\pi^{\theta}_{0:N}$ .
>
> We thank the reviewer for the comment and agree that the reparameterization trick needs further clarification. In response, we added additional details to the updated version of the paper..

---

> > ### Comment · Reviewer_gx83 · 2025-04-01
> >
> > Hi authors,
> >
> > All my concerns are resolved. Some of them are my misunderstanding of the papers, I apologize for it. I have updated my score.
> >
> > I suggest adding discussions on the reparametrizations and gradient backpropagation to make the computation footprint clear.

---

> > > ### Author Response · Authors · 2025-04-02
> > >
> > > We thank the reviewer for their quick answer and for adjusting the score. As the reviewer suggested, we added additional details to the updated version of the paper.

---

### Official Review · Reviewer_TJZx · 2025-03-13

**Overall Recommendation:** 4

**Summary:**

This paper introduces DIME, a novel online RL algorithm using diffusion policies. The key innovation lies in proposing a new method for maximizing the entropy of a diffusion policy with more rigorous theoretical justification. The authors establish a lower bound for policy entropy and derive a practical diffusion RL algorithm based on this foundation. The algorithm can be interpreted as minimizing the KL divergence between the marginal distributions of the reverse process defined by the diffusion model and the forward diffusion process by adding noise to the maximum entropy optimal action distribution at each diffusion step. Extensive experiments conducted across multiple benchmark environments demonstrate superior sample efficiency and converged performance compared to baseline methods in most cases.

**Claims And Evidence:**

Yes, the experimental results clearly demonstrate the effectiveness of the proposed algorithm.

**Essential References Not Discussed:**

I did not find any.

**Experimental Designs Or Analyses:**

Yes, the authors’ experimental design is reasonable, and they conducted extensive experiments with multiple random seeds to mitigate statistical uncertainty.

**Methods And Evaluation Criteria:**

Yes, the proposed algorithm effectively maximizes the entropy of the diffusion policy.

**Other Comments Or Suggestions:**

In Equation 26, the $Q_\theta$ should be $Q_\phi$.

**Other Strengths And Weaknesses:**

### Strengths

1. The writing of the paper is very clear, with well-structured theoretical derivations that progress logically.

2. This paper addresses a crucial issue in applying diffusion models to online RL: policy entropy estimation. It presents a novel solution distinct from previous works and enables non-Gaussian exploration patterns.

3. The experiments in this paper are comprehensive, demonstrating superior performance compared to baseline methods like DIPO.

### Weaknesses

1. The experimental section lacks comparisons with some of the latest diffusion-based online RL methods, such as QVPO and DACER.

2. The authors employ distributional RL in DIME but do not specify whether the baseline methods also use it. Additionally, they do not provide corresponding ablation studies, making it unclear how much improvement is attributed to the use of distributional RL.

**Questions For Authors:**

See weaknesses.

**Relation To Broader Scientific Literature:**

This paper is closely related to diffusion-based offline and online RL, as well as standard maximum entropy RL.

**Theoretical Claims:**

No, most of the proofs align with standard derivations in maximum entropy RL, but I have not thoroughly verified their correctness.

---

> ### Author Rebuttal · Authors · 2025-03-31
>
> We thank the reviewer for taking the time to review our work and for the many helpful comments and suggestions. We hope the following replies address the questions and concerns raised.
>
> ---
>
> > The experimental section lacks comparisons with some of the latest diffusion-based online RL methods, such as QVPO and DACER.
> >
> We thank the reviewer for the comments and their time. We have run additional experiments comparing DIME against the most recent diffusion-based RL baselines QVPO and DACER. We report the results over 10 seeds here: https://imgur.com/a/3h8hMkB . DIME converges faster to a similar performance as DACER on the Ant-v3 task and significantly outperforms both methods on the Humanoid-v3 task. Please note that the results reported in the DACER paper differ from our results. In their work, the X-axis corresponds to the number of updates, which is not the number of environment interactions. To obtain the number of environment interactions, the X-Axis would need to be multiplied by 20.
>
> ---
>
> > The authors employ distributional RL in DIME but do not specify whether the baseline methods also use it. Additionally, they do not provide corresponding ablation studies, making it unclear how much improvement is attributed to the use of distributional RL.
> >
> We thank the reviewer for this comment. We agree that the descriptions of the baselines are a bit short. We have extended the updated version of the paper. Except for BRO and BRO (Fast), all methods do not employ distributional Q by default. A comparison to a distributional variant of CrossQ has been done in Fig. 1c)- d) already.
>
> We agree that additional ablations strengthen the paper. Therefore, we have conducted additional experiments that provide additional evidence: https://imgur.com/a/N6sUqxQ. DIME with distributional Q improves performance slightly, but DIME w/o distributional Q performs on par or better than BRO, and outperforms other baselines, especially the diffusion-based methods. Please note that BRO employs quantile distributional RL.
>
> Additionally, we analyzed the effect of distributional Q on the Ant-v3 and Humanoid-v3 tasks. More concretely, we compare DIME against variants of diffusion-based baselines with distributional Q, i.e., we compare against DACER, Diffusion-QL with distributional Q, and Consistency-AC with distributional Q. Please note that DACER employs distributional Q by default. Similarly, we compare DIME without distributional Q against the diffusion-based variants without distributional Q. The results can be seen here: https://imgur.com/a/xBGkPl7
> DIME w/o distributional performs on par with baselines on the Ant-v3 task and performs significantly better on the humanoid-v3 task. DIME with distributional Q converges slightly faster to the same end performance as DACER on the Ant-v3 but significantly outperforms on the Humanoid-v3 task.
>
> ---
>
> > In Equation $Q_\theta$ should be $Q_\phi$
> >
>
> We thank the reviewer for pointing out this typo.
>
> We thank the reviewer for raising these questions, which led us to conduct additional experiments. We will include those results in the future version.

---

### Official Review · Reviewer_Xhj7 · 2025-03-14

**Overall Recommendation:** 3

**Summary:**

This paper combines maximum entropy reinforcement learning with the diffusion model, leading to a coherent algorithm that addresses several critical problems that exist in previous methods. The main contribution of this paper is that it derives a tractable lower bound of the entropy of the action distribution for diffusion-based policies. Based on this foundation, this paper transforms the traditional MaxEnt RL framework to accommodate diffusion-based policies, which enhances the exploration ability as compared to other diffusion-based algorithms. The empirical evaluations are conducted on several complex continuous control tasks, and the proposed method, DIME, does demonstrate improved performance.

## update after rebuttal
Most of my concerns are resolved by the rebuttal, except for the cost of back-propagation through the diffusion chain. I would encourage the authors to consider addressing this limitation since this would be necessary to apply DIME to large-scale applications such as generative model fine-tuning. I will maintain my positive evaluation of this paper.

**Claims And Evidence:**

Yes, main claims in this paper are well supported by theory or empirical evidence.

**Essential References Not Discussed:**

Most of the relevant literature are properly cited. However, in the field of offline RL, there are a few articles that also employs diffusion-based policies, such as Diffusion-DICE [2], DAC [3], and QGPO [4]. The authors are encouraged to include a discussion about these works.

[1] Mao, L., Xu, H., Zhan, X., Zhang, W.,  & Zhang, A. Diffusion-dice: In-sample diffusion guidance for offline reinforcement learning.

[2] Fang, L., Liu, R., Zhang, J., Wang,  W., & Jing, B. Y. Diffusion Actor-Critic: Formulating  Constrained Policy Iteration as Diffusion Noise Regression for Offline  Reinforcement Learning.

[3] Lu, C., Chen, H., Chen, J., Su, H.,  Li, C., & Zhu, J. Contrastive energy prediction for  exact energy-guided diffusion sampling in offline reinforcement learning.

**Experimental Designs Or Analyses:**

The authors designed two types of experiments to demonstrate the effectiveness of their design:
- Three **ablation studies** examined the role of the reward scaling parameter $\alpha$, the impact of diffusion steps, and the improvement brought by diffusion policies compared to Gaussian policies.
- **Performance comparisons** with other reinforcement learning algorithms.

Issues:
- The comparison with baseline methods may not be strictly fair. For non-diffusion methods, hyper-parameters such as the target entropy, the width of the critic networks are not aligned with DIME. It would be better to include additional experiments to ablate the effect of these parameters to demonstrate the benefit of diffusion compared to Gaussian policies.

**Methods And Evaluation Criteria:**

Yes. The authors test DIME on several tasks with high-dimentional observations and actions selected from three benchmarks. These are widely acknowledged benchmarks for online reinforcemenet learning.

**Other Comments Or Suggestions:**

Eq. 33, the RHS of the equation should be $\frac{\pi_{0:N}(a^{0:N}|s)}{\pi_{1:N|0}(a^{1:N}|s, a^0)}$.

**Other Strengths And Weaknesses:**

One of the main weaknesses of this paper is that the policy improvement step (Eq. 27) seems to require gradient back-propagation through the whole diffusion generation path, similar to Diffusion-QL. In practice this incurs significant memory cost and computational overhead.

**Questions For Authors:**

In Section 4.4, the authors mentioned that the diffusion coefficient $\beta$ is optimized alongwith the RL process. Could the authors provide illustrations about the learned $\beta$ at the end of RL training?

**Relation To Broader Scientific Literature:**

This article is related to two major fields: MaxEnt RL and diffusion-based policies.

MaxEnt RL augments the traditional RL objective with the policy entropy to promote exploration. The most famous algorithm in MaxEnt RL is SAC [1]. On the other hand, Diffusion-based policies have stronger expressive power compared to those based on simple distributions such as Gaussian or Dirac distribution. Although the first attempt towards diffusion-based policies merged in offline RL [2], diffusion policies have so far been widely applied in online RL (e.g. QSM [3], DACER [4], and QVPO[5]). However, combing diffusion policies with MaxEnt RL is challenging because it is generally difficult to complute the log-probability of the diffusion policy analytically, and DIME offers a nice approach to this through a tractable lower bound of the entropy term.

[1] Haarnoja, T., Zhou, A., Abbeel, P., and Levine, S. Soft  actor-critic: Off-policy maximum entropy deep reinforcement learning  with a stochastic actor. ICML. 2018.

[2] Wang, Z., Hunt, J. J., and Zhou, M. Diffusion policies as an  expressive policy class for offline reinforcement learning. ICLR. 2023

[3] Psenka, M., Escontrela, A., Abbeel, P., and Ma, Y. Learning a  diffusion model policy from rewards via q-score matching. ICML. 2024.

[4] Wang, Y., Wang, L., Jiang, Y., Zou, W., Liu, T., Song, X., Wang, W., Xiao,  L., WU, J., Duan, J., and Li, S. E. Diffusion actor-critic with entropy  regulator. NeurIPS. 2024.

[5] Ding, S., Hu, K., Zhang, Z., Ren, K., Zhang, W., Yu, J., Wang, J., and Shi, Y. Diffusion-based reinforcement learning via q-weighted  variational policy optimization. NeurIPS. 2024.

**Theoretical Claims:**

Yes, I have checked the theory. The main point of the theory is the lower bound of the entropy of the action distribution, based on which the rest largely follows the theory established by SAC.

---

> ### Author Rebuttal · Authors · 2025-03-31
>
> We thank the reviewer for taking the time to review our work and for the many helpful comments and suggestions. We hope the following replies address the questions and concerns raised.
>
> ---
>
> > The comparison with baseline methods may not be strictly fair. For non-diffusion methods, hyper-parameters such as the target entropy, the width of the critic networks are not aligned with DIME. It would be better to include additional experiments to ablate the effect of these parameters to demonstrate the benefit of diffusion compared to Gaussian policies.
> >
>
> We thank the reviewer for the comments and for their time. We agree that additional experiments that analyze various architecture designs are interesting. However, we would like to note that  Fig. 1c-d show the learning curves for DIME and a Gaussian policy under the same setup with the same critic and actor networks for 3M environment interactions until convergence. These experiments showcase the benefit of using a diffusion policy over a Gaussian policy.
>
> Additionally, we have tuned CrossQ in the benchmarks to obtain the best performance and have chosen the same or comparable networks as DIME. Please note, while DIME uses simple MLPs for the critic, BRO uses more complex network structures (BRONET) with residual connections, boosting the performance [1] and potentially providing a benefit against DIME. We believe that more sophisticated network structures might even help boosting DIME’s performance and consider this research as a promising field for future work.  Please note that the target entropy values for DIME need to vary compared to a Gaussian-based policy, as we consider the sum of log ratios of distributions (Eq.15) instead of the log probability of a single Gaussian as entropy. These instances can therefore not be directly compared and need other values as a target.
>
> [1] Naumann et al. Bigger, regularized, optimistic. Neurips 2024.
>
> ---
>
> > Most of the relevant literature are properly cited. However, in the field of offline RL, there are a few articles that also employs diffusion-based policies, such as Diffusion-DICE [2], DAC [3], and QGPO [4]. The authors are encouraged to include a discussion about these works.
> >
>
> We thank the reviewer for pointing us to recent works on diffusion-based offline RL. We will extend the related work accordingly in the final version. Could the reviewer please provide a reference to QGPO, as this is not listed in the review?
>
> ---
>
> > One of the main weaknesses of this paper is that the policy improvement step (Eq. 27) seems to require gradient back-propagation through the whole diffusion generation path, similar to Diffusion-QL. In practice this incurs significant memory cost and computational overhead.
> >
>
> We thank the reviewer for sharing their concerns and agree that gradient back-propagation through the chain can be limiting. However, in practice, this only becomes problematic in very high dimensions, such as when fine-tuning large image-models [1]. In fact, most recent works on diffusion-based sampling/inference employ equal update schemes while scaling to problems with 1000+ dimensions using hundreds of diffusion steps [2,3,4].
>
> However, if one wants to avoid having to compute gradients through the simulation, one could use the adjoint ODE for computing gradients, which describes the evolution of the gradient of the objective with respect to the parameters (see [5]). Another alternative is to use different loss functions such as the log-variance loss [6] or adjoint matching [1].
>
> [1] Domingo-Enrich, Carles, et al. "Adjoint matching: Fine-tuning flow and diffusion generative models with memoryless stochastic optimal control." ICLR 2025
>
> [2] Vargas, Francisco, Will Grathwohl, and Arnaud Doucet. "Denoising diffusion samplers." ICLR 2023
>
> [3] Vargas, Francisco, et al. "Transport meets variational inference: Controlled monte carlo diffusions." ICLR 2024
>
> [4] Blessing, Denis, et al. "Underdamped diffusion bridges with applications to sampling." ICLR 2025
>
> [5] Li, Xuechen, et al. "Scalable gradients for stochastic differential equations." AISTATS 2020
>
> [6] Richter, Lorenz, and Julius Berner. "Improved sampling via learned diffusions." ICLR 2024
>
> ---
>
> > Eq. 33, the RHS of the equation should be …
> >
>
> We thank the reviewer for pointing out this typo, which we have fixed.
>
> ---
>
> > In Section 4.4, the authors mentioned that the diffusion coefficient $\beta$ is optimized alongwith the RL process. Could the authors provide illustrations about the learned $\beta$  at the end of RL training?
> >
> As requested, we provide the adaptation of the $\beta$ value during the training under this link: https://imgur.com/a/ASqSHCe . We have extended the updated version of the paper with the corresponding figure.

---

### Official Review · Reviewer_Etdt · 2025-03-14

**Overall Recommendation:** 4

**Summary:**

Authors have introduced the idea of using diffusion-based policy in maximum entropy RL. To tackle the issue of intractable entropy calculation of using a diffusion-based policy, the authors derive a computationally tractable lower bound on the maximum entropy objective. Through extensive experiments, the proposed algorithm Diffusion-based Maximum Entropy RL (DIME) is shown to outperform existing diffusion-based methods and state-of-the-art non-diffusion based methods.

**Claims And Evidence:**

Main claims made in the submission are generally supported by clear evidence.

**Essential References Not Discussed:**

The paper provides a strong background on diffusion-based RL and MaxEnt RL literature, citing the key relevant works.

**Experimental Designs Or Analyses:**

Some ablations is reported on just one environment. It would be better to include all. For example, for hyper-parameter alpha and diffusion steps, are the optimal values the same across different environments? How sensitive are they?
No explicit reports on computation complexity (for example, wall-clock time, GPU, memory). As you claimed that DIME is computational

**Methods And Evaluation Criteria:**

Yes, the proposed algorithm DIME is tested on various standard RL environments with 10 seeds and compared against different baselines. Ablation study on important hyper-parameters is also provided for analysis.

**Other Comments Or Suggestions:**

See sections above.

**Other Strengths And Weaknesses:**

See sections above.

**Questions For Authors:**

As mentioned above, for hyper-parameter alpha and diffusion steps, are the optimal values the same across different environments? How sensitive are they? For results in figure 3, what is the algha value/diffusion steps used?
Also, any computational complexity analysis (wall-clock time, GPU)?

**Relation To Broader Scientific Literature:**

The paper extends the MaxEnt-RL literature by introducing diffusion policies as a more expressive policy class over traditional Gaussian policies.  Unlike previous diffusion-based methods, this paper uniquely addresses entropy maximization explicitly through a theoretically justified lower-bound entropy objective.

**Theoretical Claims:**

Theoretical part looks ok to me. Though theoretical part is a very important part of the paper. Math is not my strength so chairs may need to rely more on others' reviews.

---

> ### Author Rebuttal · Authors · 2025-03-31
>
> We thank the reviewer for taking the time to review our work and for the many helpful comments and suggestions. We hope the following replies address the questions and concerns raised.
>
> ---
>
> > Some ablations is reported on just one environment. It would be better to include all. For example, for hyper-parameter alpha and diffusion steps, are the optimal values the same across different environments? How sensitive are they? For results in figure 3, what is the algha value/diffusion steps used? No explicit reports on computation complexity (for example, wall-clock time, GPU, memory).
> >
>
> We thank the reviewer for their comments and time. The sensitivity of the crucial hyperparameters $\alpha$ and the number of diffusion steps are analyzed in Fig.1a-b and Fig.2a.
> Fig.1 shows the sensitivity to $\alpha$. As expected, too high and too low $\alpha$ values lead to worse performance, while $\alpha=0.001$ performs best. This trend aligns with previous maximum entropy RL papers [1], where tuning $\alpha$ per environment is suggested. However, inspired by previous works [2] we propose optimizing $\alpha$ for a target entropy (Section 4.4), eliminating the need for tuning $\alpha$ for each environment. Please note that the target entropy depends on the action dimension of the considered environment and does not need to be tuned (please see App. E).
>
> Similarly, the number of diffusion steps is task-depended, but generally, we do not observe a decrease in performance with more diffusion steps (Fig. 3a). However, a more complex task might need more diffusion steps. Our experiments found that using 16 diffusion steps performs well for all tasks.
>
> While we agree that evaluating a set of hyperparameters on all tasks leads to the optimal performance, this would also require a large amount of compute. We have therefore opted to optimize for the hyperparameters during learning, as done for $\alpha$, or fix the number of diffusion steps to a reasonable value, which works well for most of the tasks.
>
> Fig. 2 b plots the wall-clock time for the whole training process (including testing the policy) for 1M environment interactions. The plot shows that the higher the number of diffusion steps is, the longer the training takes, which makes sense given the increased computation. However, with an average training time of around 4.5 h with 16 diffusion steps DIME runs reasonably fast compared to high-performing baselines like BRO that need an average training time of 8.5h [3] on the same hardware.
>
> [1] Haarnoja, T. et al. Soft Actor Critic. ICML 2018
>
> [2] Haarnoja, T et al. Soft actor-critic algorithms and applications. 2018
>
> [3] Naumann et al. Bigger, regularized, optimistic. Neurips 2024.

---

### Decision · Program_Chairs · 2025-05-01

**Decision:**

Accept (poster)

**Comment:**

This paper presents DIME, a novel approach leveraging diffusion-based policies within maximum entropy reinforcement learning. It proposes a tractable objective, i.e. the entropy lower bound. All reviewers recommended acceptance, a decision I endorse.